



# Revealing a significant isotopic offset between plant water and its sources using a global meta-analysis.

Javier de la Casa[1], Adrià Barbeta[2], Asun Rodríguez-Uña[1], Lisa Wingate[3], Jérôme Ogée[3], Teresa E. Gimeno [1, 4]

[1] Basque Centre for Climate Change (BC3), 48940 Leioa, Spain
[2] BEECA, Department of Evolutionary Biology, Ecology and Environmental Sciences, Universitat de Barcelona, Barcelona, Catalonia, Spain.
[3] INRAE, Bordeaux Sciences Agro, UMR ISPA, 33140 Villenave d'Ornon, France
[4] IKERBASQUE, Basque Foundation for Science, 48008 Bilbao, Spain

*Correspondence to*: Javier de la Casa (delacasa.snchz@gmail.com)

**Abstract.** Isotope-based approaches to study plant water sources rely on the assumption that root water uptake and within-plant water transport are non-fractionating processes. However, a growing number of studies have reported offsets between plant and source water stable isotope composition, for a wide range of ecosystems. These isotopic offsets can result in the erroneous attribution of source water used by plants and potential overestimations of groundwater uptake by the vegetation. We conducted a global meta-analysis to quantify the magnitude of these plant-source water isotopic offsets and explore whether their variability could be explained by either biotic or abiotic factors. Our database compiled 112 studies, spanning arctic to tropical biomes that reported the dual water isotope composition ($\delta^2$H and $\delta^{18}$O) of plant (stem) and source water, including soil water. We calculated $^2$H offsets in two ways: a line conditioned excess (LC-excess) that describes the $^2$H deviation from the local meteoric water line, and a soil water line conditioned excess (SW-excess), that describes the deviation from the soil water line, for each sampling campaign within each study. We tested for the effects of climate (air temperature and soil water content), soil class and plant traits (growth form, leaf habit, wood density and parenchyma fraction and mycorrhizal habit) on LC-excess and SW-excess. Globally, stem water was more depleted in $^2$H than soil water (SW-excess < 0) by 3.02 ± 0.65 ‰. In 95% of the cases where SW-excess was negative, LC-excess was negative, indicating that the uptake of water from mobile pools was unlikely to explain the observed soil-plant water isotopic offsets. SW-excess was more negative in cold and wet sites, whereas it was more positive in warm sites. Soil class and plant traits did not have any significant effect on SW-excess. The climatic effects on SW-excess suggest that methodological artefacts are unlikely to be the sole cause of observed isotopic offsets. Instead, our results support the idea that these offsets are caused by isotopic heterogeneity within plant stems whose relative importance will depend on soil and plant water status and evaporative demand. Our results would imply that plant-source water isotopic offsets may lead to inaccuracies when using the isotopic composition of bulk stem water as a proxy to infer plant water sources.

## 1 Introduction

For decades, it has been suggested that the stable isotope composition of water (i.e. its $^2$H/$^1$H and $^{18}$O/$^{16}$O ratios, usually reported in ‰ VSMOW as $\delta^2$H and $\delta^{18}$O, respectively) in plant stems could be used to identify the origin of root water uptake and plant transpiration (Ehleringer and Dawson, 1992). Comparison of the isotopic composition of plant water with that of its potential sources has served to infer groundwater uptake in arid and semi-arid environments (e.g. Thorburn et al., 1995; Yin et al., 2015), to characterize seasonal shifts in root water uptake across the soil profile (Eggemeyer et al., 2009;



Schwendenmann et al., 2015) or to unveil the use of alternative water sources such as dew or fog (Burgess and Dawson, 2004). In the past decade, methodological advances such as novel statistical tools (Stock et al., 2018) and high throughput of samples using laser-based instruments (Martín-Gómez et al., 2015), have allowed for significant increases in the spatio-temporal resolution of water isotope datasets that can be used to infer plant water sources. Recently, several meta-analyses have
compiled such studies and found that water stored in the unsaturated zone is likely the main water source accessed by vegetation (Amin et al., 2020), with notable exceptions in arid and semi-arid environments where groundwater forms a significant contribution to the plant water budget (Barbeta and Peñuelas, 2017; Evaristo and McDonnell, 2017).

The attribution of plant water sources from the analysis of water stable isotope composition relies heavily on the assumption that the isotopic composition of plant stem water reflects that of its source. This is because root water uptake is generally
considered a non-fractionating process, so that plant and source water should have the same isotopic composition (Ehleringer and Dawson, 1992). This lack of fractionation was supported experimentally more than 80 years ago for plants grown hydroponically (Washburn, E. W., & Smith, 1934; Zimmermann U, Ehhalt D, 1967) and ever since, numerous published field studies have reported plant water isotope compositions that correspond well to a mixture of ecologically relevant potential water sources (e.g. Brunel et al., 1997; Liu et al., 2019; Rong et al., 2011; Schwendenmann et al., 2015). However, it was also
shown that isotopic offsets between plant and soil water could be found in some plants adapted to growing in xeric and saline environments (Ellsworth and Williams, 2007; Lin and Sternberg, 1993). More recently, an isotopic offset between plant stem water and pot soil water has been identified in various glasshouse experiments with non-halophytic and non-xerophytic plant species (Vargas et al., 2017; Barbeta et al. 2020) and another study that showed a symbiotic association with mycorrhizal fungi can also enhance this isotopic offset (Poca et al., 2019). Early studies suggested that these isotopic offsets resulted from an
isotopic fractionation caused by root morphological adaptations to xeric or saline environments that forced the water flow through the symplastic (cell-to-cell transport through walls and membranes) rather than the apoplastic (extracellular) pathway (Ellsworth and Williams, 2007; Poca et al., 2019). However, in the past decade, many studies have reported similar isotopic offsets between plant and source water in various biomes, including plants typical of temperate and humid ecosystems in the field (Barbeta et al., 2019; Brooks et al., 2010; Brum et al., 2019; Carrière et al., 2020; De Deurwaerder et al., 2018; Evaristo
et al., 2016; Geris et al., 2015; Tetzlaff et al., 2021), in addition to controlled experiments (Barbeta et al., 2020a; Vargas et al., 2017). Much of this literature overlooks such plant-source water isotopic offsets (Anderegg et al., 2013; Muñoz-Villers et al., 2018), whereas other studies acknowledge these offsets and attribute them to either missing water sources not sampled in the field (Bowling et al., 2017) or to the isotopic separation of water pools in the soil (Brooks et al., 2010; Vargas et al., 2017). Importantly, failing to identify the cause of these plant-source isotopic offsets can lead to biased estimates of plant water use
from potential sources, including an over-estimation of groundwater use by vegetation (Barbeta et al., 2019; Zuecco et al., 2020).

The first mechanism proposed to explain plant-source isotopic offsets was that isotopic fractionation occurred during the cell-to-cell transport of water molecules through water channels (aquaporins) in the root endodermis, which would discriminate
against $^2$H (Ellsworth and Williams, 2007; Mamonov et al., 2007; Poca et al., 2019). More recently, a series of studies have identified other plausible causes. For example, methodological artifacts associated with water extraction or isotope analysis protocols could cause apparent fractionation. It is known that the water isotopic composition of plant and soil water samples measured using laser-based instruments can be biased due to contamination of the absorption spectra by organic compounds (Brand, 2010; Schmidt et al., 2012; West et al., 2010). For this reason, spectral corrections have been developed for these
laser-based instruments (Martín-Gómez et al., 2015) and reproducible results have been demonstrated for soil and plant samples measured with laser- and mass-spectrometers (Bowling et al. 2017; Barbeta et al., 2020b). Potential issues associated with water extraction protocols are more complicated to harmonize, particularly for cryogenic vacuum distillation (CVD,





Orlowski et al., 2018). Besides parameters inherent to the CVD protocol (mainly extraction time, temperature and vacuum line pressure), soil texture, cation exchange capacity and organic matter content have been shown to affect the isotopic composition of extracted soil water (Chen et al., 2021; Orlowski et al., 2018). Alternatives to CVD exist for soil samples, such as water extraction with suction lysimeters (e.g Carrière et al., 2020) or online measurements of liquid-vapor equilibration (Dubbert et al., 2013), but CVD is still, by far, the most common methodology (Amin et al., 2020). The isotopic composition of stem water could also be altered following CVD, as hydrogen exchange between water and cellulose during extraction should cause a systematic, and potentially significant, depletion of the extracted water in $^2$H (Chen et al., 2020). Apparent fractionation could also be caused by within-stem isotopic heterogeneity created by isotopic surface effects in soil (Chen et al., 2016a) and stem (Barbeta et al., 2020b) water pools. In studies where sap water was extracted more directly [taking advantage of positive root pressure (Zhao et al. 2016); or using mechanical squeezing, using a Scholander pressure chamber (Geißler et al., 2019; Magh et al., 2020; Zuecco et al., 2020) or directional centrifugation along the stem main axis, using a Cavitron apparatus (Barbeta et al. 2020b)], no significant isotopic offsets were found between sap and source water. In addition, the CVD-extracted water remaining in non-conductive tissues, as well as bulk stem water, have both been shown to be depleted in $^2$H relative to sap water (Barbeta et al., 2020b, Zuecco et al., 2020). These recent findings would suggest that isotopic offsets would be more likely when water contained in non-conductive tissues constituted a larger proportion of bulk stem water, for example under water stress or in species with few small xylem vessels. Most often, detailed measurements of such anatomical traits are only available for discrete study sites (e.g. Cosme et al., 2017), but fortunately other proxies of anatomical traits like wood density or parenchyma fraction are more widely available (Chave et al., 2009; Morris et al., 2018). In addition, isotopic enrichment of stem water above source water can result from evaporative enrichment caused by water loss through the bark under hot and dry conditions (Martín-Gómez et al., 2017). Importantly, none of these mechanisms are mutually exclusive: for example, Barbeta et al. (2020a) found that the isotopic offset between plant and source water in potted saplings disappeared under water-limited conditions, and argued that this was caused by a combination of surface isotopic effects in soil and stem water pools with evaporative enrichment of stem water as a result of the reduction in sap flow. A systematic characterization of the global patterns of these plant-source isotopic offsets and their correlations with abiotic and biotic drivers would be the first step towards identifying their most likely underlying mechanisms.

Scattered evidence across the literature suggests that the mismatch in isotopic composition between plant and source water could be more widespread than previously assumed, but we still lack a systematic quantification of its extent and variability. In this study, isotopic offsets between plant and source water are quantified by means of the line conditioned excess (LC-excess, Landwehr and Coplen, 2006), and analogous metrics (Barbeta et al., 2019). A negative LC-excess indicates that the plant is accessing water that has undergone evaporative enrichment, for example shallow soil water (Zhao et al., 2020a), Because water stored in the soil is the most likely water source for the vast majority of plants (Amin et al., 2020). To detect isotopic mismatches between plant and source water, we should compute the δ$^2$H offset between plant water and its corresponding soil water line (SW-excess, Barbeta et al., 2019), in addition to the LC-excess. Here, we calculated LC-excess and SW-excess values from a compilation of 112 studies reporting the dual water isotopic composition of plant (stem) and source waters and analysed their relationship with environmental and climatic conditions (air temperature and soil moisture content). In addition, we also assessed the influence of ecologically relevant factors, including mycorrhizal habit and plant functional traits, mediating nutrient and water-use strategies (Chen et al., 2016b; Flo et al., 2021), as well as the comparison between taxonomic groups (angiosperms vs. gymnosperms) with known distinct hydraulic architecture and functioning (Johnson et al., 2012). Our aim was to quantify potential isotopic offsets between plant and source water and their relationship with biotic and abiotic drivers. We sought to test whether these offsets were likely driven by methodological, biological or abiotic factors. We expected that in the case where these offsets were the result of methodological artifacts, we would not find any correlation between the magnitude of this offset and environmental or biological variables, whereas significant correlations



between the offset and certain drivers would help identify possible underlying mechanisms. Following previous arguments (Barbeta et al. 2019, Poca et al. 2019), we hypothesised that plant-source isotopic offsets: (*i*) would not be restricted to xeric and saline environments and instead would be found across all biomes, also these offsets (*ii*) would increase in plants with a higher fraction of stem water in non-conductive tissues (i.e. under low water availability and in species with higher wood

density and parenchyma fraction) and (*iii*) would be more likely in plants that are known to establish mutualistic relationships with mycorrhizal fungi.

## 2 Materials and methods

### 2.1 Plant and source water isotopic composition


To compile a dataset reporting the dual water isotopic composition for plant and soil samples, we first pooled and reviewed studies included in three previous meta-analyses (Amin et al., 2020; Barbeta and Peñuelas, 2017; Evaristo and McDonnell, 2017). We then added studies obtained after a bibliographic search for peer-reviewed papers published after 2016 on Scopus, Web of Science and Google Scholar. The search was performed in December 2020 using the terms: (water AND isotop*) 

AND (dual OR (hydrogen AND oxygen)) AND (plant OR tree OR vegetat*). After screening the title and abstract, we selected studies that reported: 1) plant (stem) and source water isotopic composition, including the soil; 2) $\delta^{18}O$ and $\delta^{2}H$ for both plant (stem) and source water and 3) sufficient simultaneous soil data ($n \geq 3$) to fit a soil water line (SWL). Our final database contained 112 studies (Table 1). For each study, the isotopic composition ($\delta^{18}O$ and $\delta^{2}H$) of plant (stem) and source water was obtained from the associated published datasets, provided by the corresponding author/s or extracted from figures in the article 

using WebPlotDigitizer (Rohatgi, 2020). Plant water included water extracted from wood cores, lignified stems and rhizomes, never leaves or other transpiring tissues and hereafter is referred to as "stem water". Source waters included: soil water, precipitation, groundwater and streamflow. Here, we refer to precipitation, groundwater and streamflow as "mobile waters". For all studies, we recorded ancillary data including information of the study site, methodology and study species. Information of the study site included: geographic location of the sampling sites (latitude, longitude and elevation), climate (mean annual 

temperature and precipitation), slope and intercept of the local meteoric water line (LMWL) and study type (experimental studies on potted plants under controlled conditions, observational studies in irrigated urban gardens or agricultural fields and observational studies under natural conditions). For those studies that did not report the slope and intercept of the LMWL, these parameters were calculated from estimates of isotopic composition of precipitation obtained from the Online Isotopes in Precipitation Calculator (OIPC3.0, Bowen, 2017; Bowen et al., 2005; IAEA/WMO, 2015). Information on the methodology 

included: soil water extraction method (suction lysimeter; direct equilibration; or vacuum distillation, including CVD, azeotropic vacuum distillation, and other similar methodologies), plant water extraction method (direct xylem water extraction, direct vapor equilibration or vacuum distillation) and instrument type used for analyses of water isotopic composition (mass spectrometer or laser spectrometer). From the 112 studies reviewed, 94 studies used vacuum distillation (mainly CVD) to extract both stem and soil water. One study used direct equilibration of liquid-vapor for both soil and stem water (Bertrand et 

al., 2014). For extraction of soil water, 10 studies combined vacuum distillation and suction lysimeters and four studies used only suction lysimeters. For stem water, 108 studies used vacuum distillation, two studies used mechanical squeezing using a Scholander pressure chamber (Geißler et al., 2019; Magh et al., 2020), one study used hand-pump suction (Jiménez-Rodríguez et al., 2019). Vacuum distillation was the most common methodology for water extraction of soil (93%) and stem water (96%). Hence, our database did not allow for a robust analysis of the potential effects of water extraction methodology on plant-source 

isotopic offsets. Information of the study species included: species name, taxonomic group (angiosperm or gymnosperm), leaf habit (deciduous, semi-deciduous or evergreen), leaf shape (broadleaved or narrow-leaved) and growth form (tree, shrub or non-woody). When available, we also recorded the sampling date (month-year) and plot within the study site. For our analyses,



we grouped all data within each study site, sampling date and plot (when available) into what we called 'campaigns'. Our initial database consisted of 555 campaigns, from 112 studies (Table 1).


**Table 1.** List of studies for which SW-excess and LC-excess were calculated with the species, country, number of sampling plots and dates. Asterisks (*) next to the number of species indicates that although several species were sampled within a study it was not possible to ascribe distinct water isotopic compositions to each species.

| ID | Reference | Species | Country | Plots | Sampling dates |
|----|-----------|---------|---------|-------|----------------|
| 1 | Anderegg et al., 2013 | 1 | USA | 1 | 3 |
| 2 | Barbeta et al., 2015 | 3 | Spain | 2 | 5 |
| 3 | Barbeta et al., 2019 | 2 | France | 1 | 6 |
| 4 | Barbeta et al., 2020a | 1 | G.E. | 2 | 1 |
| 5 | Berry et al., 2014 | 2 | USA | 2 | 2 |
| 6 | Bertrand et al., 2014 | 7* | Switzerland | 1 | 1 |
| 7 | Beyer et al., 2016 | 5* | Namibia | 1 | 1 |
| 8 | Bijoor et al., 2012 | 15 | USA | 6 | 1 |
| 9 | Bodé et al., 2020 | 31* | Tanzania | 4 | 1 |
| 10 | Boutton et al., 1999 | 6 | USA | 1 | 1 |
| 11 | Bowling et al., 2017 | 2* | USA | 1 | 3 |
| 12 | Brandes et al., 2007 | 1 | Germany | 1 | 1 |
| 13 | Brinkmann et al., 2018 | 2 | Switzerland | 1 | 15 |
| 14 | Brooks et al., 2010 | 1 | USA | 1 | 3 |
| 15 | Brunel et al., 1995 | 1 | Australia | 1 | 3 |
| 16 | Brunel et al., 1997 | 1 | Camerun | 1 | 1 |
| 17 | Brum et al., 2017 | 15* | Brasil | 1 | 1 |
| 18 | Brum et al., 2019 | 12 | Brasil | 1 | 1 |
| 19 | Cao et al., 2018 | 1 | China | 1 | 1 |
| 20 | Carrière et al., 2020 | 3 | France | 1 | 1 |
| 21 | Chi et al., 2019 | 3 | China | 1 | 4 |
| 22 | Cramer et al., 1999 | 2* | Australia | 1 | 1 |
| 23 | De Deurwaerder et al., 2018 | 2 | French Guiana | 2 | 1 |
| 24 | Dong et al., 2020 | 1 | China | 4 | 1 |
| 25 | Dubbert et al., 2019 | 2 | Portugal | 1 | 8 |
| 26 | Dudley et al., 2018 | 1 | New Zealand | 1 | 11 |
| 27 | Dwivedi et al., 2020 | 2 | USA | 1 | 2 |
| 28 | Eggemeyer et al., 2009 | 4 | USA | 1 | 9 |
| 29 | Estrada-Medina et al., 2013 | 2 | Mexico | 1 | 2 |
| 30 | Evaristo et al., 2016 | 2 | Puerto Rico | 2 | 2 |
| 31 | Feikema et al., 2010 | 2 | Australia | 1 | 2 |
| 32 | Gaines et al., 2016 | 9 | USA | 1 | 1 |
| 33 | Geris et al., 2015 | 1 | UK | 2 | 1 |
| 34 | Geris et al., 2017 | 2 | UK | 3 | 3 |
| 35 | Gierke et al., 2016 | 1 | USA | 1 | 2 |





| 36 | Goldsmith et al., 2012 | 6* | Mexico | 2 | 2 |
|----|------------------------|-----|-------------|---|---|
| 37 | Gómez-Navarro et al., 2019 | 6* | USA | 1 | 3 |
| 38 | Grossiord et al., 2017 | 1 | USA | 1 | 3 |
| 39 | Guo et al., 2016 | 1 | China | 1 | 4 |
| 40 | Hartsough et al., 2008 | 1 | Mexico | 1 | 2 |
| 41 | Hervé-Fernández et al., 2016 | 4* | Chile | 2 | 2 |
| 42 | Holland et al., 2006 | 1 | Australia | 1 | 1 |
| 43 | Jespersen et al., 2018 | 4 | USA | 1 | 4 |
| 44 | Jia et al., 2018 | 1 | China | 1 | 3 |
| 45 | Jones et al., 2020 | 1 | Australia | 4 | 2 |
| 46 | Knighton et al., 2020 | 2 | USA | 8 | 3 |
| 47 | Kulmatiski et al., 2006 | 2 | USA | 1 | 2 |
| 48 | Leng et al., 2013 | 3 | China | 1 | 1 |
| 49 | Li et al., 2020 | 1 | China | 3 | 5 |
| 50 | Liu et al., 2014 | 3 | China | 1 | 1 |
| 51 | Liu et al., 2019a | 1 | China | 1 | 4 |
| 52 | Liu et al., 2018 | 4 | China | 1 | 1 |
| 53 | Liu et al., 2019b | 2 | China | 1 | 1 |
| 54 | Liu et al., 2020 | 1 | China | 1 | 1 |
| 55 | Luo et al., 2019 | 1 | China | 1 | 4 |
| 56 | Ma and Song, 2016 | 1 | China | 1 | 2 |
| 57 | Marttila et al., 2018 | 1 | New Zealand | 1 | 1 |
| 58 | McCole and Stern, 2007 | 1 | USA | 1 | 2 |
| 59 | McCutcheon et al., 2017 | 11 | USA | 3 | 6 |
| 60 | Mensforth et al., 1994 | 1 | Australia | 4 | 2 |
| 61 | Moore et al., 2016 | 1 | USA | 1 | 3 |
| 62 | Muñoz-Villers et al., 2018 | 4 | Mexico | 1 | 2 |
| 63 | Muñoz-Villers et al., 2020 | 3 | Mexico | 1 | 3 |
| 64 | Nehemy et al., 2020 | 1 | G.E. | 1 | 2 |
| 65 | Newberry et al., 2017 | 1 | G.E. | 2 | 1 |
| 66 | Nie et al., 2011 | 5 | China | 1 | 1 |
| 67 | Ohte et al., 2003 | 3 | China | 3 | 1 |
| 68 | Poca et al., 2019 | 1 | G.E. | 1 | 1 |
| 69 | Qian et al., 2017b | 3 | China | 1 | 1 |
| 70 | Qian et al., 2017a | 1 | China | 1 | 4 |
| 71 | Ripullone et al., 2020 | 3 | Italy | 1 | 1 |
| 72 | Rong et al., 2011 | 5 | China | 1 | 2 |
| 73 | Rose et al., 2003 | 2 | USA | 1 | 3 |
| 74 | Rossatto et al., 2012 | 20* | Brasil | 1 | 1 |
| 75 | Schulze et al., 1996 | 8 | Argentina | 7 | 1 |
| 76 | Schwendenmann, 2015 | 7 | Panama | 1 | 1 |
| 77 | Schwendenmann and Jost, 2019 | 11 | Panama | 3 | 1 |
| 78 | Schwendenmann, 2019 | 2 | Indonesia | 1 | 5 |





| 79 | Simonin et al., 2014 | 5* | USA | 1 | 2 |
|---|---|---|---|---|---|
| 80 | Snelgrove et al., 2020 | 4 | Canada | 4 | 6 |
| 81 | Snyder and Williams, 2000 | 3 | USA | 1 | 1 |
| 82 | Song et al., 2014 | 1 | China | 1 | 1 |
| 83 | Song et al., 2016 | 1 | China | 1 | 7 |
| 84 | Sun et al., 2019 | 3* | China | 2 | 1 |
| 85 | Swaffer et al., 2014 | 2 | Australia | 1 | 1 |
| 86 | Voltas et al., 2015 | 1 | Spain | 1 | 3 |
| 87 | Wang et al., 2017 | 3 | China | 3 | 3 |
| 88 | Wang et al., 2019 | 2 | China | 1 | 4 |
| 89 | Wei et al., 2013 | 1 | China | 1 | 1 |
| 90 | West et al., 2007 | 2 | USA | 1 | 1 |
| 91 | Wu et al., 2016a | 4 | China | 1 | 6 |
| 92 | Wu et al., 2018 | 2 | China | 2 | 1 |
| 93 | Wu et al., 2016b | 4 | China | 1 | 1 |
| 94 | Yang et al., 2015 | 3* | China | 1 | 28 |
| 95 | Yin et al., 2015 | 1 | China | 1 | 1 |
| 96 | Zhang et al., 2011 | 1 | China | 1 | 2 |
| 97 | Zhao et al., 2019 | 3 | China | 1 | 2 |
| 98 | Zhao et al., 2020a | 2 | China | 1 | 1 |
| 99 | Zhao et al., 2020b | 2 | China | 2 | 1 |
| 100 | Zhou et al., 2019 | 1 | China | 3 | 1 |
| 101 | Zhu et al., 2016a | 1 | China | 3 | 4 |
| 102 | Zhu et al., 2016b | 2 | China | 1 | 1 |

### 2.2 Calculation of SW-excess and LC-excess

We fitted a soil water line (SWL) for isotopic composition of soil water from samples collected within each campaign (observations within a study, sampling date and plot) according to Eq (1) (Sprenger et al., 2016):

$$\delta^2 H_s = a_s \times \delta^{18}O_s - b_s ,\qquad (1)$$

where $\delta^2 H_s$ and $\delta^{18}O_s$ correspond to soil water samples from various depths, locations or pots (in the case of glasshouse experiments); within a site (or plot), sampling date and study. Fitted parameters $a_s$ and $b_s$ are the slope and intercept of the SWL. Parameters $a_s$ and $b_s$ were calculated only for those campaigns where the linear relationship between $\delta^2 H_s$ and $\delta^{18}O_s$ were significant ($P < 0.05$). In this step we discarded 133 campaigns, corresponding to nine studies (Geißler et al., 2019; Huang and Zhang, 2015; Liu et al., 2011; Lovelock et al., 2017; Magh et al., 2020; McKeon et al., 2006; Saha et al., 2015; Su et al., 2020; Twining et al., 2006). Next, we estimated the difference in δ²H between each plant water sample and its corresponding soil water line (SW-excess) according to Eq (2) (Barbeta et al., 2019):

$$SW - excess = \delta^2 H_p - a_s \times \delta^{18}O_p - b_s,\qquad (2)$$





where $\delta^2H_p$ and $\delta^{18}O_p$ denote the isotopic composition of individual plant water samples. We calculated a value of SW-excess for each plant sample and averaged values within species and campaigns. We discarded 14 campaigns, because they lacked concurrent observations for plant water. In addition, of the remaining 103 studies, all but one (Jiménez-Rodríguez et al., 2019) reported isotopic composition of bulk plant water and thus this study was not included in the end. The final number of studies included was 102 with 407 campaigns and 197 species.

To measure how well defined SW-excess is for a given species and campaign, we computed the standard error of the mean of SW-excess ($\sigma_{SW-ex}$) (Taylor, 1997):

$$\sigma_{SW-ex} = \sqrt{\left(\sigma_{^2H_p}\right)^2 + \left(a_s \times \sigma_{^{18}O_p}\right)^2 + \left(\sigma_{a_s} \times \sigma_{^{18}O_p}\right)^2 + \left(\sigma_{b_s}\right)^2}, \qquad (3)$$

where $\sigma_{a_s}$ and $\sigma_{b_s}$ are the standard errors of the slope and intercept of the SWL, respectively and $\sigma_{^2H_p}$ and $\sigma_{^{18}O_p}$ are the standard errors of the mean (per species and campaign) of $\delta^2H_p$ and $\delta^{18}O_p$, respectively. To characterize how dispersed SW-excess was for a given campaign, we also calculated the variance of SW-excess:

$$Var(SW-excess) = Var\left(\delta^2H_p\right) + a_s^2 Var\left(\delta^{18}O_p\right) - 2a_s Cov(\delta^2H_p\delta^{18}O_p), \qquad (4)$$

where $Var()$ and $Cov()$ denote the variance and covariance of variables. For studies reporting one value of $\delta^2H_p$ and $\delta^{18}O_p$ per species or campaign, we estimated their variance from the means of all other calculated $\sigma_{^2H_p}$, $\sigma_{^{18}O_p}$ and $cov_{^2H_p,^{18}O_p}$.

Similar statistics were derived for the line conditioned excess (LC-excess) according to Eq (5) (Landwehr and Coplen, 2006):

$$LC-excess = 2\delta^2H_p - a_L \times \delta^{18}O_p - b_L, \qquad (5)$$

where $a_L$ and $b_L$ are the slope and intercept of the corresponding LMWL. We calculated a value of LC-excess for each plant sample (only for observational studies) and then averaged values within species and campaigns. The standard error of the mean ($\sigma_{LC-ex}$) and variance [Var(LC-excess)] of LC-excess were calculated as in Eqs (3) and (4), but assuming that $\sigma_{a_L}$ and $\sigma_{b_L}$ were zero. For each campaign, we considered that either the LC- or SW-excess were different from zero when its estimate plus or minus its standard error was greater or smaller than zero.

### 2.3 Climatic, environmental and biological data

Climatic and environmental data were extracted from the ERA5-Land Copernicus data service (Hersbach et al., 2019), downloaded from the Copernicus Climate Change Service (C3S) Climate Data Store. For each study site and sampling date, we extracted: mean monthly and annual air temperature at 2 m above the surface, monthly and annual total precipitation, monthly and annual potential evapotranspiration, mean monthly and annual soil volumetric water content (VWC) at four depth intervals (0-7, 7-28, 28-100 and 100-289 cm) and average soil water content for the upper 100 cm (calculated from soil VWC of the upper soil layers: 0-7, 7-28 and 28-100 cm). In addition, for each study site, we extracted soil class from the ERA5 database according to the FAO/UNESCO Digital Soil Map of the World (FAO and UNESCO, 2003). This database classifies the soil within each ~10 km pixel into seven soil classes: coarse, medium, medium fine, fine, very fine, organic and tropical organic. Finally, for each plant species we obtained average values of wood density [Chave et al. (2009), available for 65





species, representing 258 out of the 656 observations], parenchyma volume fraction [Morris et al. (2018), available for 26 species of angiosperms representing 150 out of the 656 observations] and mycorrhizal habit (i.e. whether a certain plant species
has been reported to establish a symbiotic association with either an arbuscular or ectomycorrhizal fungi, both types of fungi or none of them [from Maherali et al. (2016), available for 118 species and hence for 408 out of the 656 observations].

**2.4 Statistical analyses**

Our final database consisted of 656 records of mean values of SW-excess and 642 of LC-excess (the LC-excess was not calculated for glasshouse studies), for 197 species and 407 campaigns gathered from 102 studies. We used linear mixed models (LMMs) to assess the effects of biotic and abiotic variables on the slope of the SWL, SW-excess and LC-excess, including study as a random factor. To assess the global prevalence of isotopic offsets between plant water and its potential sources, first, we ran LMMs without fixed factors (null models). Next, in the fixed part of the model, we included the following potential
explanatory variables: mean monthly air temperature, annual potential evapotranspiration, monthly and annual precipitation, mean monthly soil VWC, soil class and methodology used for analyses of water isotopic composition for the slope of the SWL, LC-excess and SW-excess; and wood density, fraction of parenchyma, leaf habit, growth form, leaf shape, mycorrhizal habit and taxonomic group for LC-excess and SW-excess. All explanatory variables were included in our LMMs in standardized form. In addition, for the LMMs assessing potential effects of plant traits measured at the species level (wood
density, parenchyma fraction and mycorrhizal habit), species identity was included as a random factor of the model, because some species were measured in multiple studies. We performed individual models for each explanatory variable and those that had significant effects were tested in combination in additive models. Estimated effects for the SWL slope, LC-excess and SW-excess were weighted by the inverse of the variance, to consider the precision of the information given by each study (Koricheva et al., 2013). In meta-analytical models, two potential sources of variation might be accounted: the random
sampling variability within each study (i.e. within-study heterogeneity) and the additional variability between studies, caused for instance by different experimental conditions (i.e. between-study heterogeneity).Thus, we calculated a heterogeneity statistical index to test the percentage of variation across studies caused by between- rather than within-study heterogeneity ($I^2$, Higgins & Thompson 2002). The 95% confidence intervals of the $I^2$ indices were 96.60–99.61%, 99.85–99.85% and 99.98–99.98% for the SWL slope, LC-excess and SW-excess, respectively, indicating that most variation corresponded to between-
study heterogeneity. We selected the LMMs including random effects (i.e. accounting for both between- and within-study heterogeneity) rather than those with only fixed effects (i.e. only accounting for within-study heterogeneity), as they fitted the data better in terms of the Akaike Information Criterion (AIC) (Burnham and Anderson, 2002). Therefore, both within and between-study heterogeneities were included in the models. In addition, to disentangle direct and indirect effects of environmental variables on the SW-excess, we ran additional mixed models. We aimed to assess whether any observed effect
on the SW-excess was caused by a preceding effect of the same variable on the SWL parameters (slope and intercept). Indeed, those linear regression parameters are used to calculate the SW-excess, and they could be potentially affected by environmental variables. Therefore, we extracted the residuals of the correlations of the SW-excess with the SWL parameters, and subsequently introduced them in a model with the relevant environmental variables. This way, only those effects that would be significant in this second model (using residuals as response variable) could be considered as direct environmental effects
on the SW-excess. All analyses were performed in R (version 4.0.3, R Core Team, 2020) using packages: *lme4* (Bates et al., 2015), *lmerTest* (Kuznetsova et al., 2017), *MuMIn* (Barton, 2009), *standardize* (Eager, 2017), *emmeans* (Searle et al., 1980) and *performance* (Lüdecke et al., 2020)





## 3 Results

### 3.1. Combined analysis of SW-excess and LC-excess

We compared SW-excess and LC-excess within species and campaigns and found that SW-excess was negative in 184 campaigns (out of 642 campaigns, glasshouse studies excluded). We found that for 95% of these campaigns (175 out of 184), LC-excess was also negative, (Figure 1). In these 175 cases, there would be a mismatch in isotopic composition between stem and source water, regardless of whether plants were taking up mobile water (precipitation, groundwater or streamflow) or water stored in the soil that had been subject to evaporation enrichment. These 175 cases were distributed across 57 of the 153

study sites with no apparent bias linked to geographical region (Figures 2 and 3). In addition, we found 12 campaigns for which both SW-excess and LC-excess were positive, likely resulting from evaporative enrichment affecting stem water, irrespective of the water source and of potential isotopic offsets.

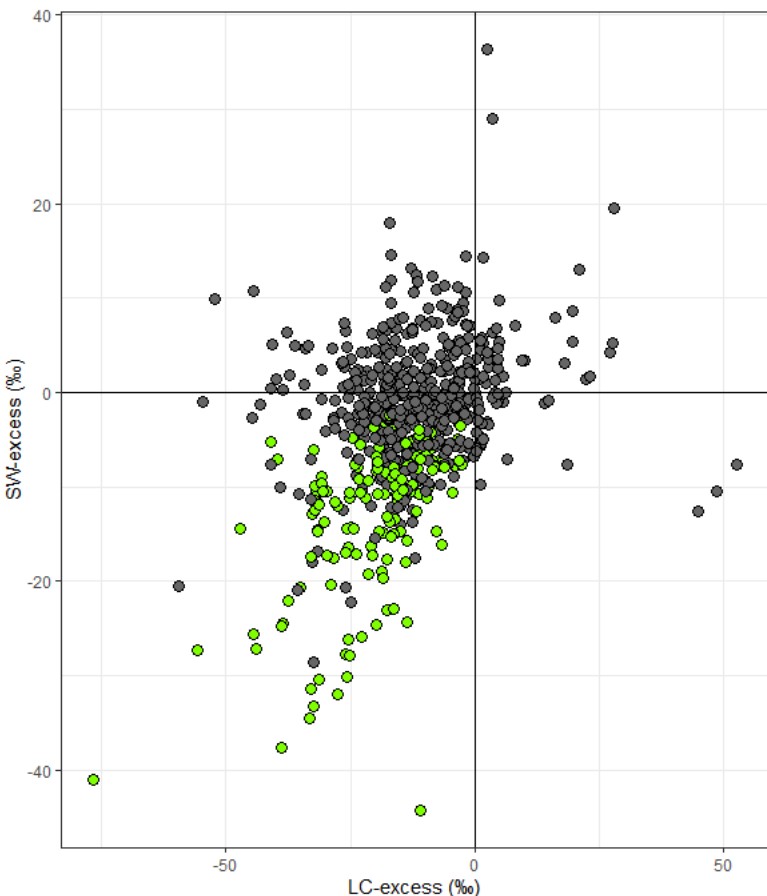

**Figure 1.** Soil water-excess (SW-excess) plotted against line conditioned-excess (LC-excess). Points are the mean values per species and campaign (observations from a given date and plot within a study). Green symbols indicate observations where both LC-excess and SW-excess plus their corresponding standard error were negative. Error bars have been omitted for clarity. Note that the scales of the x and y axes are different.






**Figure 2.** Mean (±SE, *n* = number of campaigns per study) soil water excess (SW-excess) for each study site. The dotted line
is the overall mean estimate of SW-excess (-3.02 ‰). Each bar corresponds to a sampling site (see ID numbers on the left and
in Table 1 for the corresponding references). Green bars depict study sites where both the line conditioned excess (LC-excess)
and SW-excess (plus their corresponding SE) were negative for at least one campaign, grey bars depict study sites where LC-
excess and/or SW-excess were not different from zero for all campaigns. Colourless bars depict glasshouse studies.

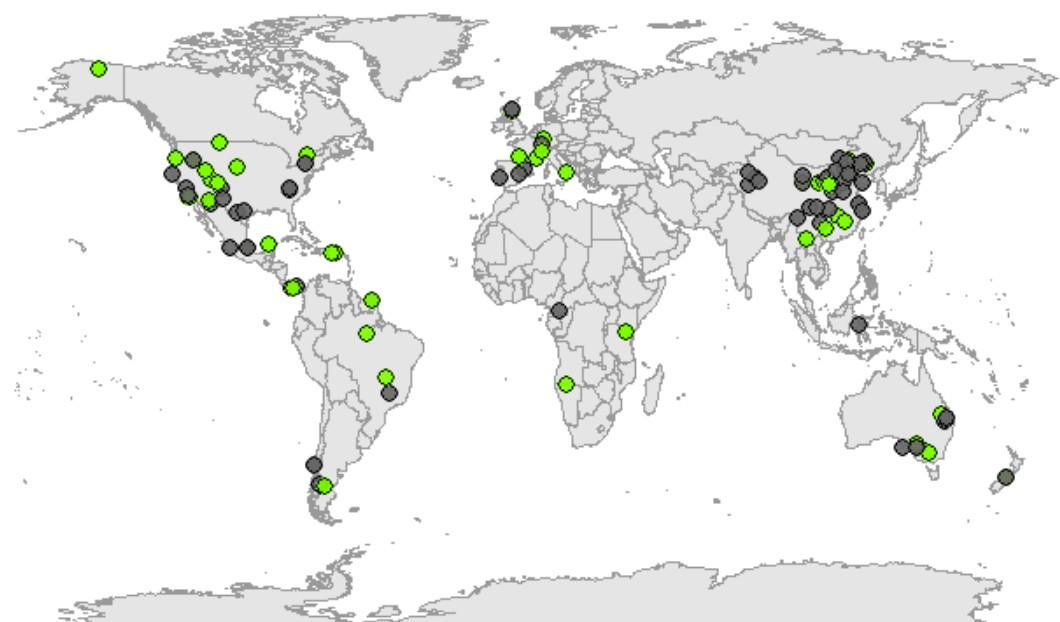

**Figure 3.** Map showing the sampling location for the study sites (data from experiments under controlled conditions excluded). Green symbols depict studies where both line conditioned excess (LC-excess) and soil water excess (SW-excess) were negative for at least one campaign.


### 3.2 Overall value and effects of abiotic variables on the slope of the soil water line (SWL)

The linear regression between $\delta^2$H and $\delta^{18}$O of soil water samples was significant for 422 of the 555 campaigns compiled. Of all campaigns for which the SWL regression was significant, the mean SWL slope was $5.52 \pm 0.17$ ($\pm$SE) and the mean

intercept was -16.0 $\pm$ 2.4 ‰. According to the results of the null LMM (Table 2) and considering the weight and the random variability across studies, the overall mean SWL slope was significantly positive and lower than that of the global meteoric water line ($P < 0.001$). The LMMs including climate variables in the fixed part of the model indicated that the SWL slope decreased (i.e. greater evaporative enrichment) with warmer temperatures ($P < 0.001$), increased with monthly and annual soil VWC of the upper soil layers ($P < 0.001$ for monthly values of 0-7, 7-28 and 28-100 cm; $P = 0.014$, 0.016 and 0.034 for annual

values of 0-7, 7-28 and 28-100 cm, respectively), with integrated soil water content ($P < 0.001$ for monthly values for soil depths 0-100 and 0-289 cm, $P = 0.024$ and $P = 0.034$ for annual values for soil depths 0-100 and 0-289 cm, respectively) and with annual ($P = 0.013$) and monthly precipitation ($P < 0.001$). We did not find any significant differences in the SWL slope among soil classes (Table S2). The methodology for measuring soil water isotopic composition (mass- vs laser-spectrometers) did not have any significant effect on the estimated SWL slope ($P = 0.327$ Table S2).


### 3.3 Overall estimates of line conditioned-excess (LC-excess) and effects of biotic and abiotic variables

We calculated 642 mean values of LC-excess, from 400 campaigns, 98 studies and 194 species (glasshouse experiments excluded). The overall mean value of LC-excess was significantly negative (-12.2 $\pm$ 1.3 ‰, $P < 0.001$, for the LMM with no

fixed effects), indicating that, overall, plant water samples fell below their corresponding LMWL in the dual isotope space. The annual potential evapotranspiration had a positive effect on LC-excess ($P < 0.001$). There were some differences in LC-excess among soil classes ($P = 0.028$): LC-excess was less negative in organic than in medium texture soils, although our database only included five observations for the soil class "organic" (Table S2). LC-excess also differed among plant types





according to mycorrhizal habit ($P = 0.014$): values of LC-excess were more negative in plants that have been shown to form

associations with both types of mycorrhizal fungi (Table S2), while no differences were found between ectomycorrhizal or

arbuscular associations ($P = 0.999$). We did not find significant differences between LC-excess values calculated from

measurements of stem water isotopic composition measured with mass- or laser-spectrometers ($P = 0.421$).

**3.4 Overall estimates of soil water excess (SW-excess) and effects of biotic and abiotic variables**

We calculated 656 mean values of SW-excess, from 407 campaigns and 197 species, using observations of stem water isotopic

composition and the slope and intercept of their corresponding SWL. The overall mean estimate of SW-excess was

significantly negative (-3.02 ± 0.65 ‰, $P < 0.001$, according to the LMM with no fixed effects), indicating that there was an

overall significant isotopic offset between stem and soil water.

We found that there was a significantly positive relationship between SW-excess and monthly air temperature ($P < 0.001$;

Figure 4a) and monthly potential evapotranspiration (PET; $P = 0.002$), and a significantly negative relationship between SW-

excess and mean monthly soil VWC of the upper soil layers (0-7, 7-28 and 28-100 cm; $P < 0.001$), but not with soil VWC

from deeper soil layers (Table S1). SW-excess was also significantly and negatively correlated with integrated soil water

content for the upper (0-100 cm) soil ($P < 0.001$; Figure 4b). Neither monthly, nor annual precipitation were significantly

correlated with SW-excess (Table S1). When assessed in combination, we found that monthly air temperature still had a

significantly positive correlation with SW-excess ($P < 0.001$), but soil water content did not ($P = 0.083$). Importantly, a more

detailed analysis of the residuals of the relationship between SW-excess and the SWL parameters (slope and intercept) revealed

that only the temperature effects had a direct effect on SW-excess (Table S3). On the other hand, the observed effects of soil

VWC on SW-excess appeared to be a consequence of the direct effect of soil VWC on the SWL slope and intercept (Table

S3).

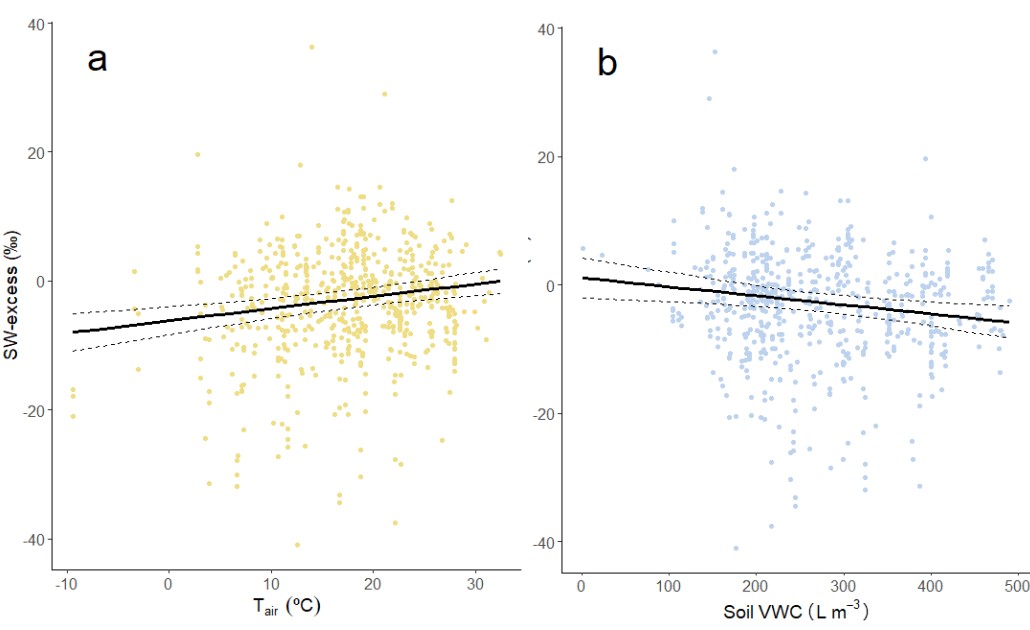






**Figure 4.** soil water excess (SW-excess) plotted against a) mean monthly air temperature ($T_{air}$) and b) monthly soil volumetric water content of the upper 100 cm. Each point is the mean SW-excess per species and campaign (observations from a given sampling date and plot/site within a study). The solid and dashed lines are the prediction and standard error of the corresponding linear mixed model, with a single predictor variable.


**Table 2**. Results ($t$ and $P$), sample size ($n$) and parameter estimates according to the linear mixed models (including the null models without any predictor variables) to assess the effects of temperature, soil volumetric water content (VWC) and potential evapotranspiration (PET) on the slope of the soil water line (SWL slope), line-conditioned excess (LC-excess) and soil water excess (SW-excess). All parameter estimates have been standardized. Only models with significant results are shown.


| Response Variable | Predictor variable | n | Estimate | Std. error | t-value | P-value |
|---|---|---|---|---|---|---|
| | Null model | 656 | 5.52 | 0.17 | 31.687 | <0.001 |
| SWL slope | Monthly air temperature | 609 | -0.32 | 0.07 | -4.58 | <0.001 |
| | Monthly precipitation | 609 | 0.45 | 0.10 | 4.44 | <0.001 |
| | Annual precipitation | 639 | 0.40 | 0.15 | 2.51 | 0.013 |
| | Monthly VWC (0-7 cm) | 609 | 0.90 | 0.10 | 8.99 | <0.001 |
| | Monthly VWC (7-28 cm) | 609 | 0.80 | 0.09 | 8.45 | <0.001 |
| | Monthly VWC (28-100 cm) | 609 | 0.60 | 0.10 | 5.54 | <0.001 |
| | Monthly soil water content (0-100 cm) | 609 | 0.72 | 0.10 | 6.77 | <0.001 |
| | Monthly soil water content (0- 289 cm) | 609 | 0.52 | 0.13 | 3.99 | <0.001 |
| | Annual VWC (0-7 cm) | 639 | 0.34 | 0.13 | 2.48 | 0.014 |
| | Annual VWC (7-28 cm) | 639 | 0.32 | 0.13 | 2.41 | 0.016 |
| | Annual VWC (28-100 cm) | 639 | 0.27 | 0.13 | 2.19 | 0.034 |
| | Annual soil water content (0-100 cm) | 639 | 0.30 | 0.13 | 2.27 | 0.024 |
| | Annual soil water content (0- 289 cm) | 639 | 0.28 | 0.13 | 2.12 | 0.034 |
| | Monthly PET | 572 | -0.33 | 0.077 | -6.75 | <0.001 |
| | Annual PET | 639 | -0.52 | 0.014 | -2.32 | 0.021 |
| | Null model | 642 | -12.230 | 1.32 | -9.02 | <0.001 |
| LC-excess | Annual PET | 632 | -2.52 | 1.00 | -2.50 | 0.013 |
| | Null model | 656 | -3.02 | 0.65 | -4.59 | <0.001 |
| SW-excess | Monthly air temperature | 609 | 1.00 | 0.32 | 3.06 | 0.002 |
| | Monthly VWC (0-7 cm) | 609 | -1.94 | 0.44 | -4.32 | <0.001 |
| | Monthly VWC (7-28 cm) | 609 | -1.61 | 0.42 | -3.80 | <0.001 |
| | Monthly VWC (28-100 cm) | 609 | -1.51 | 0.48 | -3.14 | 0.002 |
| | Monthly soil water content (0-100 cm) | 609 | -1.69 | 0.47 | -3.55 | <0.001 |
| | Monthly soil water content (0- 289 cm) | 609 | -1.25 | 0.54 | -2.29 | 0.022 |





| | | | | | |
|---|---|---|---|---|---|
| Monthly PET | 609 | 1.00 | 0.33 | 3.02 | 0.002 |

According to our results, the mean SW-excess did not differ among soil classes (Table S2). We did not find any significant difference between plant groups (Table S2): mean values of SW-excess did not differ between angiosperms and gymnosperms ($P = 0.73$), nor among growth forms (trees, shrubs and non-woody plants, $P = 0.07$), or leaf habit (deciduous, evergreen or semi-deciduous, $P = 0.63$) or leaf shape (broad or narrow, $P = 0.51$). Also, we did not find significant differences among plant groups according to their presumed mycorrhizal habit ($P = 0.64$). For those species for which we had estimates of wood density and/or parenchyma fraction, the LMMs (including species identity in the random part of the model) did not reveal any significant relationship of any of these wood anatomical variables with SW-excess (Table S1). Mean ($\pm$ SE) SW-excess values estimated from studies using either mass- or laser-spectrometers to measure stem water isotopic composition were both significantly negative: -2.1 $\pm$ 0.9 ‰ and -5.0 $\pm$ 1.1 ‰, for mass- and laser-spectrometers, respectively, abut there was a significant difference between instrument types ($P = 0.048$). Similarly, the type of instrument used to measure soil water isotopic composition also had a significant effect on SW-excess ($P = 0.015$, -2.05 $\pm$ 0.95 and -5.03 $\pm$ 1.11, for mass- and laser-spectrometers, respectively)

## 4. Discussion

Our meta-analysis revealed that the isotopic composition of stem water does not generally overlap with that of the corresponding soil water in the dual-isotope space (average SW-excess: -3.02 $\pm$ 0.65 ‰). The isotopic depletion of stem water relative to its source was originally thought to be restricted to arid or saline environments (e.g. Ellsworth and Williams, 2007; Lin and Sternberg, 1993). However, we show here that sites depicting significantly negative SW-excess (i.e. where plant water is depleted with respect to its most likely source: the soil) are more ubiquitous, and span temperate, boreal and tropical ecosystems. The combined analysis of SW-excess and LC-excess showed that for the majority of cases where SW-excess was negative (95%), LC-excess was also negative. This result indicates that plant water uptake from sources other than soil water that have not undergone evaporative enrichment (such as groundwater), cannot explain the observed mismatch in isotopic composition between plant and soil water. Instead, our results call into question the general assumption that plant water faithfully reflects the isotopic composition of its source.

We compiled 112 studies reporting dual isotopic composition of plant water and its sources and we estimated values of LC-excess and SW-excess for 102 of them. These estimates were widely distributed across the globe, encompassing boreal, tropical, temperate, Mediterranean, arid and semi-arid ecosystems. However, overall there is a literature bias towards data collection in temperate forests, consequently these ecosystems were overrepresented in our database. In contrast, observations from tropical ecosystems were the scarcest, in line with the observations of previous meta-analyses of stable isotope data of plant water and its sources (Amin et al., 2020; Barbeta and Peñuelas, 2017; Evaristo and McDonnell, 2017). Here, we aimed to partially overcome the limited climatic variability represented by biome type and geographic location by incorporating seasonal climatic variability, specific to each study site when available. To do so, we gathered monthly values of air temperature and soil water availability for each study plot and sampling campaign, encompassing a large breadth of climatic values spanning from -10 up to 35 °C for mean monthly air temperature and from 1 to 50% for soil VWC. Our analyses from this data compilation agreed with predictions from classic theory, for example, we found that, globally, the slope of the soil water line (SWL) is smaller as temperature increases and water availability reduces, because of increased evaporative enrichment (Craig and Gordon, 1965; Sprenger et al., 2016)





At the global scale, we found positive effects of monthly air temperature and negative effects of soil VWC on the SW-excess. One of the main results from our analysis was that the SW-excess was clearly most negative in cooler and wetter environments. This result is in agreement with recent observations from an array of boreal forests, where significant offsets (i.e. negative SW-excess) were found in all study sites, with the two coldest sites depicting the most negative values of SW-excess (Tetzlaff et al., 2021). The SW-excess is calculated with the slope and intercept of the soil water line, but in turn, those parameters correlate with soil VWC and air temperature (see above). Therefore, we run a subsequent analysis of the residuals to tease apart direct and indirect environmental effects on SW-excess (Table S3). This analysis revealed that the negative effect of soil VWC on the SW-excess was mediated by the variability in the parameters of the SWL. In moist sites, the slope of the SWL was steeper and the intercept was larger, which resulted in more negative estimates of SW-excess. On the other hand, air temperature appeared to affect SW-excess more directly (Figure 4a). In cold sites, the effect of low soil water evaporative enrichment could have resulted in steeper SWL slopes, and hence more negative SW-excess (see Eq. 2). Meanwhile, on the opposite end of the temperature range, warm temperatures could be causing greater evaporative enrichment of stem water (Martín-Gómez et al., 2017), and hence partially or completely compensate the negative values of SW-excess (Barbeta et al. 2019). Taken together, these results show that isotopic offsets between plant water and its sources are largest in cold and wet places and suggest that besides evaporative enrichment, other temperature-sensitive processes could be causing these offsets in the field, such as transport or water exchange among pools within the stem.

Isotopic mismatches between sources and plant water have been identified in glasshouse and field experiments (e.g. Tetzlaff et al., 2021; Barbeta et al., 2020a; Chen et al., 2020; Vargas et al., 2017). Previously, these have been attributed to fractionation processes occurring along the soil-plant-atmosphere continuum, mostly related to hydrogen isotopes. Our meta-analysis confirms this pattern at the global scale but cannot pinpoint a definitive mechanistic explanation A recent study suggested that methodological artifacts related to $^2$H exchange with cellulose during cryogenic vacuum extraction could be at the origin of these negative SW-excess (Chen et al., 2020), at least in studies where plant stem water was extracted using CVD. Following the latter mechanistic explanation, the relative depletion in $^2$H of stem water should be associated with stem water content: plants with a lower stem water content should show more negative SW-excess values (Chen et al. 2020). If this was the case, one would expect that plants growing in drier sites should have lower stem water content and thus more negative SW-excess. This was not what we found here, and in fact drier sites tended to have less negative (smaller) SW-excess values. In the database compiled here, cryogenic vacuum distillation (CVD) was the most common methodology used for extraction of both stem and soil water, as in Amin et al., 2020. Additional recent studies under controlled conditions also suggest that during CVD, isotopic exchange both within the soil and the stem could cause apparent isotopic fractionation (Adams et al., 2020; Chen et al., 2021; Orlowski et al., 2018). However, our meta-analysis did not allow to test for the effects of extraction methodology on plant-source isotopic offsets due to the paucity of studies applying alternative methodologies to CVD, since alternative methodologies based on centrifugation (Barbeta et al., 2020b) or low-suction (Geißler et al., 2019; Magh et al., 2020; Zuecco et al., 2021) have only emerged recently. To help identify apparent fractionation caused by artifacts associated with CVD, future studies applying these novel methodologies should consider combining them with analyses of cryogenically extracted water from a concurrent subset of their samples (Geißler et al., 2019; Marshall et al., 2020). In addition, isotopic offsets between plant water and its sources are often attributed to soil properties underlying methodological artifacts, particularly also during CVD. Soil properties that can affect water isotopic composition measured following CVD include organic matter, texture, and cationic exchange capacity (Adams et al., 2020; Araguás-Araguás et al., 1995; Chen et al., 2021). In our meta-analysis, we compiled all the soil properties provided in the studies revised, but there were large inconsistencies across studies in the type of data used to describe soil properties. Hence, we opted to use soil classes derived from the Digital Soil Map of the World (FAO and UNESCO, 2003), downloaded from the ERA5 service (Hersbach et al., 2019). Here, we did



not find significant differences among soil classes in either the slope of the SWL, LC- or SW-excess. However, we
acknowledge that the soil classes used here may not be representative of the actual soil properties at each study site, due to
their coarse spatial resolution (∼10 km). In the future, it would be desirable that studies analysing soil water isotopic
composition systematically reported at least the following soil properties: soil texture (preferably by providing percentages of
sand, silt and clay), cationic exchange capacity and organic matter or total carbon; instead of merely stating the soil type or
texture. Finally, our study does not completely discard potential biases in water isotopic composition associated to the type
of instrument used for measuring water isotopic composition (mass- vs. laser-spectrometers). However, instrument type cannot
explain the negative overall estimate of SW-excess, as estimates from either type were significantly negative . , Overall, our
results showed that plant-source water isotopic offsets depict significant relationships with climatic drivers and suggest that
methodological artifacts associated to isotopic measurements and cryogenic vacuum extractions are highly unlikely to be the
sole mechanisms explaining the observed source-stem water isotopic offset.


The combined analyses of LC-excess and SW-excess can help identify the type of ecosystems where we could expect larger
biases on the attribution of plant water sources from water isotopic composition. For example, in cold and/or very wet climates,
where soil water is subject to very little evaporative enrichment and the slopes of LWML and SWL are similar, when neither
LC-excess nor SW-excess were different from zero, variations in plant water isotopic composition would likely track that of
its most likely source: precipitation water (e.g. Geris et al., 2015). In arid or semi-arid ecosystems where deep-rooted vegetation
has access to groundwater, when SW-excess is different from zero, but LC-excess is not, we would infer that the vegetation
was taking up groundwater that had not undergone evaporative enrichment (e.g. Miller et al., 2010). Conversely, in temperate
ecosystems, when SW-excess was zero, whereas LC-excess was negative, most likely, isotopic composition of plant water
would also match the most likely source: water stored in the upper soil, subject to evaporative enrichment (Brinkmann et al.,
2018). However, in hot climates, evaporative enrichment could affect plant water isotopic composition too, irrespective of the
water source and of potential isotopic offsets (Martín-Gómez et al., 2017) and partially or completely compensate for potential
isotopic offsets. In any case, the various possible mechanisms underlying isotopic mismatches are not mutually exclusive and
multiple mechanisms exerting opposing effects could coexist, but these can only be disentangled in experiments under
controlled conditions (Barbeta et al., 2020a; Chen et al., 2020; Vargas et al., 2017). Still, in field studies, potential errors in
the attribution of plant water sources could be avoided, or at least identified, by including analyses of both LC-excess and SW-
excess. Yet, here, we emphasize that null values for either SW-excess and/or LC-excess might not necessarily imply absence
of offsets in isotopic composition between plant water and its sources, since these offsets can be masked by mechanisms with
opposite effects acting simultaneously (Barbeta et al., 2020a).

We expected differences among plant groups in plant-source isotopic offsets based on their anatomical traits and mycorrhizal
partnership. For example, water in non-conducting tissues has been shown to be depleted in $^{2}$H with respect to sap water (Zhao
et al., 2016, Barbeta et al. 2020b) and, thus, a greater fraction of water stored in non-conductive stem compartments could
cause larger isotopic offsets between plant and source water. Our analyses did not reveal any significant difference in either
LC-excess or SW-excess among plant groups according to their evolutionary history and hydraulic strategy (angiosperms vs.
gymnosperms), growth form (trees, shrubs or non-woody), leaf habit or morphology. Our dataset, however, did not encompass
a balanced representation of all plant groups, for example, nearly three quarters (141 out of 197) of the species included in our
meta-analysis were trees, whereas less than 20% of our observations corresponded to non-woody species (36 out of 197).
There were some differences among plant groups according to presumed mycorrhizal partnership on LC-excess. Arbuscular
mycorrhizal associations have been hypothesised to cause isotopic fractionation during root water uptake (Poca et al., 2019)
and therefore we expected larger isotopic offsets in plants forming associations with arbuscular mycorrhizae, but our results
did not support this hypothesis. We found that LC-excess, but not SW-excess, was more negative for plants that have been





shown to form mycorrhizal associations with either arbuscular or ectomycorrhizal fungi. Mycorrhizal associations are beneficial for the host plant because they increase nutrient and water availability for the plant and in return the host plant supplies carbohydrates to their mycorrhizal partner (Antunes and Koyama, 2017). Given the carbon costs of these associations

for the plant, to maximise their investment return, we would expect that plants forming mycorrhizal associations would allocate larger proportions of their root and fungal hyphal biomass to the shallower soil layers, where nutrient concentrations are higher (Esteban and Robert, 2001). This could explain the more negative LC-excess observed in plants forming mycorrhizal associations, as their main water source would be shallow soil water (subject to evaporative enrichment), instead of deep mobile water pools.


We also explored correlations between plant-source isotopic offsets and two wood anatomical traits: wood density and parenchyma fraction, at least in angiosperms (Morris et al., 2018). If isotopic heterogeneities within stem water pools underlie isotopic offsets (e.g. Barbeta et al., 2020b; Zhao et al., 2016), then we should observe larger isotopic offsets in species where sap water constitutes a smaller fraction of total stem water, i.e. species with narrower conduits, higher parenchyma fraction

and denser wood. Our results did not agree with this prediction and suggest that anatomical traits might not be good predictors of plant-source isotopic offsets. Nonetheless, our results do not discard isotopic heterogeneity within the stem as a plausible mechanism driving observed offsets. Isotopic heterogeneity between water pools within the stem can still result in isotopic offsets between bulk stem water and source water, but the extent of this offset would be determined by the actual plant relative water content at the time of measurement (Barbeta et al., 2020a; Chen et al., 2020), more than wood anatomical traits alone.


**5. Conclusions**

We calculated LC-excess and SW-excess from more than a hundred studies distributed globally and found that overall, plant water did not isotopically match the considered source waters. This isotopic offset was largest in cold and wet sites, where

plant water plotted below and/or to the right of source water in the dual isotope space, whereas plant water generally plotted closer to the soil water line in hot climates. Our results call into question the long-standing assumption that plant water isotopic composition faithfully reflects that of its source. Based on the recent literature, this does not seem to be the case for $\delta^2H$, at least. The significant correlations found between the magnitude of these plant-source isotopic offsets with temperature and with soil moisture suggest that these offsets are unlikely caused by purely methodological artifacts. However, the ultimate

mechanisms driving these isotopic offsets and their ecological significance can only be unveiled with experiments under controlled conditions. The results from our meta-analysis suggest that these experiments should include comparisons of contrasting soil properties, plant species with varying wood traits and encompass gradients of plant relative water content and storage. These experiments would shed light on the most plausible mechanisms underlying these isotopic offsets and contribute to avoid erroneous attributions of source water from analyses of water isotopic composition.


**Code availability**

The code used for all statistical analyses is available upon request.

**Data availability**

All the data will be made available on a publicly accessible online repository upon acceptance for publication.

**Author contribution**



AB, TEG, LW and JO conceived the idea and designed the study. JdlC performed the literature review and collected the data. JdlC and ARU performed the statistical analyses. JdlC and TEG wrote the first manuscript draft. All authors contributed to the

writing.

**Competing interests**

The authors declare that they have no conflict of interest.

**Acknowledgements**

We thank all the authors that kindly shared their published data and responded to our queries. Thanks to Javier Porras for his contribution to data collection and to Dominic Roye for his assistance during compilation of climatic data. Funding was provided by the Regional Government of the Basque Country (Programa Investigación Básica y Aplicada, grant: PIBA-2019-

105) and by the Spanish Ministry of Science (grant PHLISCO PID2019-107817RB-I00).

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
