# Peer review of "Isotopic offsets between bulk plant water and its sources are larger in cool and wet environments"

_Hydrology and Earth System Sciences, 2021_

## Author Response (AR1)

In this document we provide a detailed point-by-point response to the comments raised by the two reviewers that evaluated our manuscript submitted to *Hydrology and Earth System Sciences* (hess-2021-333), in addition to a non-anonymous comment. Reviewer comments are in Calibri font. In this response letter, line (L) numbers refer to the revised version without track changes.

**Reviewer 1**

**RC: This manuscript presents a straightforward, comprehensive analysis of dual-isotope offsets of plant water from soil water and precipitation. I can imagine this work being of interest and I commend the much hard work and careful thought that went into this manuscript. However, there are major issues that need to first be addressed.**

AC: We thank this reviewer for their positive evaluation of our manuscript and more importantly for the time invested and the constructive comments.

**RC: The paper claims that "Globally, stem water was more depleted in 2H than soil water (SWexcess< 0)", but the study finds that "SW-excess was negative in 184 campaigns (out of 642 campaigns)". Thus, SWexcess values below zero are actually globally rare. When we look at Figure 2, which shows +/-1SE, it is apparent (e.g., imagine a doubling of the error bars to represent an approximation of a 95% CI) that very few sites would have mean SWexcess values significantly below zero. The presentation of the discussion, abstract, and conclusions are framed around a claim that is not entirely consistent with the findings.**

AC: In the previous version of our manuscript we had not succeeded at transmitting our main findings while being truthful to the results of the statistical model. In this revised version, we have changed important sections of the results, discussion and conclusions following the comments by this reviewer and those in the same line by the second reviewer. Below, and in line with these comments, this reviewer points out that we had incorrectly stated that our calculated negative SW-excess was "ubiquitous". As pointed out by this reviewer, this would have led the reader to interpret that the significantly negative SW-excess was present across all studies. Instead, we found that significantly negative estimates of SW-excess have been measured in many different types of ecosystems across the globe (but not all), regardless of the prevailing climate. We have now clarified this issue in this revised version by changing the word "ubiquitous" for "widespread" and by stating: "Our meta-analysis revealed that the isotopic composition of plant water did not always faithfully reflect that of its most likely source and this was evident from results from many different types of biomes. The isotopic composition of stem water varied substantially in size and direction of deviation from soil water, but on average was slightly lower than soil water".

**RC: While I realize that the authors are referring to the mean SWexcess of -3.02 +/- 0.65 permil, this is a case where the average is not especially representative of the global behavior, but it is instead driven by outliers (again, see Figure 2). What is the median SWexcess? The authors need to also explain how the "+/- 0.65" was calculated, because it is unclear whether this uncertainty value only reflects the variation among the campaigns; does it also include the error from the calculation itself (eq. 3)? The manuscript needs to include a more nuanced interpretation of the findings, recognizing the wide range in values observed rather than over-relying on the mean value.**

AC: Our global estimate of SW-excess is not merely the calculated average of all SW-excess. This value (-3.02‰) is the significant estimate (*P*-value < 0.05), and its corresponding standard error (± 0.65‰), of the null model, which considers the random variability across studies and is also weighted by the sample size of each study. This is now briefly indicated in the abstract: "(P < 0.05 according to estimates of our linear mixed model and weighted by sample size within studies)". Calculated values of SW-excess depicted a normal distribution, with a mean (±SE) of -3.56 ± 0.33‰, a median of -2.58‰ and with the lower and upper limits of the 95% confidence interval being -4.22 and -2.9‰, respectively, hence also suggesting that overall there was a significantly negative SW-excess.

**RC: The closing statement of the abstract, "Our results would imply that plant-source water isotopic offsets may lead to inaccuracies when using the isotopic composition of bulk stem water as a proxy to infer plant water sources' ' still remains true even if only 184 campaigns support it. Perhaps more importantly, the inconsistency and non-ubiquity of negative SW excess values allows the authors to find the climatic effects, which may hint at ways to anticipate or predict plant-source water isotopic offsets.**

AC: We thank this reviewer for this constructive comment. We have edited the last part of our abstract (L33) which now reads: "Contrary to previous expectations, we argue that these potential biases are not restricted to saline or arid environments but should extend to many other ecosystems, notably those from wet and cold environments."

**RC: Lines 30-31: It is speculative to suggest in the abstract that these findings "support the idea that these offsets are caused by isotopic heterogeneity within plant stems". No data were used to directly test this and thus this statement has the potential to mislead readers. I recognize that the authors use their data to argue that they can rule out alternative explanations, and I know that this message is one made and supported in other works by members of this research team, but partially ruling out alternatives does not automatically lead to support for one specific explanation. If the authors choose to keep this line, it should be appropriately framed, for example, as prefaced with a phrase that explains their logic, such as "Because of a lack of alternative explanations, we suggest that these data support". In my opinion, the abstract is more robust without this sentence at all, as the prior sentence is the one that can be more robustly defended.**

AC: We agree that this statement could have come across as speculative and therefore, we have removed this sentence from the abstract, also in line with a similar suggestion by reviewer 2.

**RC: Lines 124-125: "We expected that in the case where these offsets were the result of methodological artifacts, we would not find any correlation between the magnitude of this offset and environmental or biological variables." I appreciate the authors lay out their forthcoming interpretation, but I do not understand the logic here. There could be methodological artefacts but also biological or environmental controls that are strong enough that they show up despite the influence of methodological artefacts. It also might help to be more specific about which methodological artefacts are being referred to here, because effects of using a mass spec vs a laser spec are directly tested and discussed later. Also, how does this relate to the statement on line 164, "our database did not allow for robust analysis of the potential effects of the water extraction methodology".**

AC: We have substituted this sentence by the following:

"We expected to find significant correlations between these offsets and environmental or biological drivers, that should help us identify possible mechanisms underlying these offsets. In contrast, a lack of significant correlations could suggest that methodological artefacts (mainly due to CVD, see Chen et al. 2020) would be more likely to be the main cause of these offsets."

This change was aimed to emphasize that we did expect to find significant correlations that would help us unveil the most plausible mechanisms underlying these offsets. In the case where we could not find such significant correlations, this latter result could hint that artefacts related to the process of cryogenic distillation would be the main cause of these offsets, as detailed in the previous paragraph (L87-98). In the methods section, we provide a detailed explanation of why potential artefacts associated to CVD could not be explicitly tested here (L158-165).

**RC: Line 141: What would happen if the analysis was restricted to a larger "n" value? The uncertainties in "a-s" and "b-s" must often be very large when n is only equal to 3. Is the number of values that are not significantly different from zero a consequence of this analysis including studies where soils were under sampled? I am not suggesting that the authors change the threshold, but also doing the analysis with a higher threshold might support useful further insights into the meaning of the findings.**

AC: Here, we gathered the soil water isotopic compositions reported from 508 campaigns, of which only four were discarded for having less than three observations. Among the remaining 504, there were 20 campaigns that only had three observations. The linear regressions for all these 20 campaigns were non-significant and therefore, although in theory our criteria allowed for estimates of the slope and intercept term with only three points, in practise, this threshold was less than four observations. Furthermore, the number of campaigns for which we estimated the slope and intercept of the SWL with less than five observations was low (<10%). Therefore, we do not believe that the limited number of observations for a very small number of campaigns could have biased our estimates. Indeed, if we ran our analyses to calculate SW-excess excluding all campaigns with less than ten observations for soil water isotopic composition, we obtain an overall estimate of SW-excess, according to the null linear mixed model, that is similar: $-3.47 \pm 0.69$‰. We have added a sentence in the Material and methods section of manuscript that provides this further information: "The number of slopes and estimates of SWL estimated with less than five observations ($n < 5$) was low (<10%), and we ran parallel analysis limiting fitting the SWL with higher observations ($n \geq 10$) and obtained similar results as with $n \geq 3$".

**RC: Line 147: Because "mobile water" is so frequently used to refer to soil water extracted by suction cup lysimeters in the isotope ecohydrology community, this term should be changed. Given that "mobile" is only used a few times throughout the paper, I suggest simply saying "precipitation, groundwater, and stream water" each time.**

AC: We thank the reviewer for this comment and we have modified the text accordingly (L27, L147,L278 & L494).

**RC: Line 176: Were the SWLs and LMWLs calculated by orthogonal least squares fitting? They should be because both the X and Y data have uncertainty, and fitting by standard least squares can artificially reduce slopes, which could have consequences for the findings.**

AC: For those campaigns for which the regression of the SWL was significant, there were no differences between slopes estimated with total- or orthogonal-least squares (Fig. A). Finally, we did not calculate the slope and intercept of the local meteoric water line (LMWL) for each study site, instead, these parameters were obtained from the corresponding studies (L150-154). We have added a sentence in the Material and methods section of manuscript that provides this further information: "The fitting method used was standard least squares fitting, orthogonal least squares fitting was tested parallelly and there were no differences in the estimation of slopes and intercepts of the SWL between both methods"

[Figure]

**Fig. A.** Comparison between slopes of the soil water line (SWL) estimated with either orthogonal-lest squares (OLS) or total-least squares (TLS). The line depicts the significant linear relationship between estimates ($P < 0.001$) with a slope not significantly different from 1 (0.92, with a 95% of 0.77-1.06).

**RC: Line 194: What are the criteria for plant and soil sampling to be considered "concurrent"?**

AC: By "concurrent", we meant that soil and plant samples had been collected during the same sampling day. We have clarified this in the text which now reads "We discarded 14 campaigns, because simultaneous observations for plant water collected on the same day were lacking"

**RC: Line 201: Is this equation missing a term for the product of sigma-as and delta18O? Also, please check your citations – it refers to a book review in Physics Today rather than the book itself. Another citation issue is on 670.**

AC: There was a typo in the manuscript (but not in the code used for data analysis). This equation is now corrected in the revised manuscript, and the appropriate citation for it has been updated.

**RC: Line 282: The logic underlying this attribution of these results to "evaporative enrichment affecting stem water" needs to be more clearly explained. Evaporative enrichment could lead to positive or negative SW excess values, depending on the slope of that evaporative enrichment. More generally, the conceptual model and assumptions that are guiding the interpretation of the LC-excess values should be more clearly stated.**

AC: We agree with this reviewer that this concept is too succinctly explained here, and this statement has been removed from the results section.

**RC: Line 304: Is this SE a pooled standard error of the by-campaign SWLs, or is it just of the variation among the individual values.**

AC: The text now clarifies that this estimated standard error is the SE of the null model.

**RC: Line 377-385: As written, this statement is not true because plant water generally (by study or by campaign) did overlap with the corresponding soil water, even if the average value is significantly different from zero. It would be more accurate to say "the isotopic composition of stem water varied substantially in size and direction of deviation from soil water, but on average was slightly lower than soil water".**

AC: We agree that our original wording would benefit from this clarification that better reflects our results. We have changed this sentence to better agree with the statistical results: "Our meta-analysis revealed that the isotopic composition of plant water did not always faithfully reflect that of its most likely source and this was evident from results from many different types of biomes. The isotopic composition of stem water varied substantially in size and direction of deviation from soil water, but on average was slightly lower than soil water"

**RC: Two sentences later, "sites depicting significantly negative SW-excess… are more ubiquitous" is also problematic because they are not ubiquitous, and "more ubiquitous" is a problematic construction because it means "found everywhere" and thus it cannot be used in a relative sense; use "widespread".**

AC: We thank this reviewer for this suggestion and we have changed the text accordingly by substituting "more ubiquitous" for "widespread" ).

**RC: That said, I also think it would be appropriate to cite other studies here (e.g., Chen et al., 2020, Barbeta et al 2020, and probably others) because this manuscript is not the first to suggest that there are isotopic mismatches between soil and water that occur outside of arid or saline environments.**

AC: The goal of the first paragraph of the discussion is to give a brief overview of what we consider are the main findings of our study. Here, we do not claim that our study is the first one to report plant-source isotopic offsets and later in this section (L387-393) we provide a

detailed discussion of our findings within the framework first presented by these studies, which we cite multiple times.

**RC: One more issue occurs in the following sentence: as written it could be read as indicating that SW-excess was negative in 95% of cases because the parenthetical "95%" should either follow "majority" or "also negative".**

AC: The parenthetical 95% now follows the word "majority", as suggested.

**RC: Line 417: I cannot follow the logic captured in the last sentence. How are transport and water exchange likely to be different in cold and wet places? This need to be appropriately laid out and explained rather than quickly inserted at the end. What is the conceptual model?**

AC: We have rephrased this last sentence which now reads: "Overall, our findings that the soil-stem isotopic offsets are generally larger in cold and wet environments, with minimal evaporative demand suggest that stem evaporative enrichment (as in Martín-Gómez et al. 2017) cannot be the only explanation".

Clearly, other processes, also sensitive to temperature and soil moisture must be causing these isotopic offsets in the field. Recent studies have shown that that stem storage water not participating directly to the transpiration stream is more depleted than sap water flowing through xylem vessels (Zhao et al. 2016; Barbeta et al. 2020). Although stem water content was generally not available in our meta-analysis, it would be reasonable to expect higher stem storage water content in cool and moist environments. Thus a high fraction of stem storage water could have lowered the $\delta^2H$ of bulk stem water compared to sap water, and thus source water, and lead to larger soil-stem isotopic offsets in cool, wet environments.

**RC: Line 427: Did drier sites have plants with lower stem water content? This would be a useful relationship to report.**

AC: This reviewer raises a very relevant point, which we had not highlighted sufficiently. In this revised version, we have added a sentence to this paragraph to emphasize the importance of reporting stem relative water content in studies analysing plant water isotopic composition:

"Measurements of stem relative water content are often collected during CVD to assess extraction efficiency (e.g. West et al. 2006), but these are rarely reported. In the future, it would be desirable that studies presenting water isotopic composition of different plant organs also report their relative water content, given the increasing recognition of the functional relevance of this plant trait (Martínez-Vilalta et al. 2019; Sapes & Sala 2021)."

**RC: Lines 462-464: I do not understand this sentence.**

AC: We have rephrased this sentence, which now reads: "Conversely, in temperate ecosystems, when SW-excess was not significantly different from zero and LC-excess was negative, combined analyses of water isotopic composition of plant and soil water would most likely reveal that plants were taking up water from the upper soil layers, where water would be subject to evaporative enrichment"

**RC: Discussion of Benettin et al 2018 (https://doi.org/10.5194/hess-22-2881-2018) is peculiarly absent. What may be problematic for the methods used here is that Benettin et al. show a hysteretic pattern that results from the fact that soil water "lines" reflect a combination of mixing and evaporative fractionation processes. If only part of the annual pattern were sampled (e.g., just a growing season), it would not necessarily be first-order linear, but instead it could be curvilinear. Of course a linear relationship can be fit to curvilinear data, but this could be problematic where extrapolation is involved (e.g., in calculation stem water SWexcess). In this manuscript, cases where the SWL is defined by points with X-axis values that are dissimilar to the stemwater X-axis values, the method used would project the line to the stem water values along a first-order-linear slope. If the X-axis values of stem water and soil water are highly different, the consequences of using a linear versus curvilinear SWL could be large.**

AC: In this revised version, we cite Benettin et al. (2018) in our discussion (L410 & L458) Our results are in line with the findings from this study as we advocate for the combined use of both LC-excess and SW-excess to help detect potential biases in the attribution of plant water sources instead of simply using the intersection of the SWL with the LMWL. We agree that in certain cases a curvilinear fit would be perhaps more accurate than a linear fit. However, Benettin et al. (2018) report that accounting for both evaporation and the variability in meteoric water isotopes along the seasons also results in a linear fit. Although we acknowledge that a curvilinear fit would have been possible in some cases of incomplete mixing, our dataset does not provide any objective and mechanism-based criteria from which we could apply the curvilinear fit with an objective criteria. Therefore, we conclude that our simpler approach of fitting exclusively first-order linear regressions and excluding the non-significant ones is the most adequate for our purpose.

**Reviewer 2**

**RC: This is an interesting study that adds to several other recent meta-analyses on the topic, by exploring the potential role of various biotic and abiotic factors in determining the offset between plant and source (in this case: soil) water stable isotope composition.**

AC: We thank this reviewer for these positive comments about our work.

**RC: I recommend that the authors reconsider their title (main contribution). As it stands, revealing the offset is not the main novelty that this study brings. Indeed, the second line of the abstract already states that many studies have reported this, and this has also been demonstrated by recent other meta-analysis studies. This study does include several really interesting novel aspects, that I would recommend the authors put the focus more on, e.g. the outcomes of the specific hypotheses that were tested, the relationships with air T and VWC, combined use of LC-excess and SW-excess (but see comment below). The abstract and conclusion could be slightly edited to highlight the novelty a bit more as well.**

AC: We appreciate this comment and we propose the following alternative title:

Large isotopic offsets between bulk plant water and its sources in cool and wet environments

**RC: My main concern relates to the approach towards the soil water data. Most studies used in the meta-analyses will have taken soil samples across a range of different depths. It is known that at many sites, the water stable isotope composition of soil water changes with depth (and the slope of the soil water lines, which is the basis of the analyses here, does as well); the variation with depth is also often used to determine proportional plant water uptake patterns of different soil depths, and Amin et al 2020 have shown that uptake occurs mainly from the upper part of the soil profile. The authors do not provide information on their approach to dealing with data from different depths, but it appears that they have not taken this into account and have used all soil water data available to derive soil water lines. The extend of the offset between the plant and actual source water could therefore be significantly over- or underestimated. It would be good if the authors provide a clear rationale for their approach and discuss the implications of this, and/or investigate how different the results might be e.g. for SW-excess of shallow and deep soil water as the source. Similarly, it is questionable that the authors currently consider VWC at a range of depths, but not the isotope composition.**

AC: In order to fit a soil water line (SWL) for a given field campaign there is no alternative but to use soil water isotopic composition from different depths. The theory leaves little room for interpretation in this matter: in the absence of any water inputs (precipitation), for any period (days to weeks) during which soil water is subject to evaporation, then soil water isotopic compositions along the entire soil profile (at all depths) will align along an evaporation line in the dual isotopic space. A poor linear fit usually results from incomplete mixing of different water inputs (precipitation events), at different depths (Bennetin et al 2018). That is why we only calculated values of SW-excess for those campaigns where the linear fit was statistically significant. In any case, the soil water line is always based on soil data from different depths. Therefore, it is not possible to derive a "SW-excess of shallow and deep soil water as the source" as this reviewer suggests, since using either deep or shallow water would not result into a linearly fitted soil water line. In addition, SW-excess is not an estimation of the soil water pool contributing to plant water but rather an estimation of the separation in the dual isotope space between soil and xylem water, irrespective of the processes underlying this separation.

**RC: Although there is only a small proportion of the studies that does not (only) report soil water from vacuum distillation, there is a lot of literature that has demonstrated that there are strong differences in the pore spaces that are sampled via vacuum distillation versus lysimeters. Those results also show that the LC-excess of soil water obtained via vacuum distillation can be very different from soil water obtained via suction lysimeters. I agree with the authors that it is beyond the scope of this study to address this here as well, but I would argue that the data of the suction lysimeters should not be included in the analyses here. Alternatively the authors could indicate which studies have used which type of data and explore whether these are outliers/affect the results.**

AC: The reviewer raises an important issue that we did take into account during our data collection, by carefully listing each time the extraction technique used to obtain soil water. Among all the studies that we retained for the calculation of SW-excess, only four used suction lysimeters (Jespersen et al. 2018, Lovelock et al. 2017, Yin et al. 2015, Zhang et al. 2011), and an extra ten studies combined cryogenic extraction and suction lysimeters (Dwivedi et al. 2020, Geris et al. 2015, Geris et al. 2017, Grossiord et al. 2017, Hervé-Fernández et al. 2016,

Li et al. 2020, Marttila et al. 2018 and Muñoz-Villers et al. 2018, Nehemy et al. 2020, Snelgrove et al. 2020). We added these references to the edited manuscript (L60). Unfortunately, in most of these studies no distinction was made between extraction techniques when presenting the data (unlike in Brooks et al. 2010, for example). Furthermore, we found that excluding the isotopic composition of soil water obtained from suction lysimeters did not have any impact on our overall results and interpretation, as there were no statistical differences between the SW-excess estimated for studies that used exclusively cryogenic vacuum distillation to extract soil water, and those that included water sampled with lysimeters ($P = 0.64$).

This is now clearly stated in the revised manuscript, by adding the following sentence in the Results section:

"Finally, for those studies that included soil water samples obtained by lysimeters, the estimated SW-excess was not significantly different from that estimated from studies extracting all soil water with cryogenic vacuum distillation (P = 0.64)."

**RC: Following on from the comment above, I suggest the authors do not refer to precipitation, stream water and groundwater as the 'mobile' water, as this is often used to describe the freely draining or large pore space (i.e. not tightly-bound) soil water as typically sampled by lysimeters.**

AC: Thank you for this comment, we have modified the text according to this suggestion and a similar one by reviewer 1 (L27, L147, L278 & L494)..

**RC: Finally, I wonder why the LC-excess for the soil water itself was not also included in the analyses to understand offsets. This could help with the interpretation of the relationship between plant LC-excess and SW-excess, and the role that climate or soil properties may have.**

AC: Here we used the joint analysis of plant LC- and SW-excess to test whether significantly negative values of SW-excess were likely to depict genuine offsets between plant water and its most likely source (the soil), or if instead these significantly negative SW-excess values would be caused by a plant accessing water pools of meteoric origin, instead of soil water. We believe that the discussion of the soil LC-excess and how it could be affected by climatic drivers would be beyond the scope of this study.

**RC: Regarding the abstract:**

**L 21: make it clear here that you are looking at offsets in plant water.**

AC: We now specify that these are "plant-source" $^2$H offsets

**RC: L 21: make it clear here that LC-excess is calculated based on the LMWL (not the GMWL)**

AC: Our text in the abstract already reads "a line conditioned excess (LC-excess) that describes the $^2$H deviation from the local meteoric water line". No changes have been performed.

**RC: L 30: I did not see direct evidence for the statement that offsets are caused by isotopic heterogeneity within plant stems. I'd leave this and the last sentence out of the abstract and keep it in the discussion as a possible explanation/implication.**

AC: Thank you for the appreciation, we have removed this sentence from the abstract as suggested by this reviewer and also by reviewer 1.

**RC: Introduction:**

**L56-59: would be easier to follow if this sentence was split. Also in L 58 change 'that showed' to 'showed that'**

AC: This sentence has been split into two. Now it reads: "More recently, an isotopic offset between plant stem water and pot soil water has been identified in various glasshouse experiments with non-halophytic and non-xerophytic plant species (Vargas et al., 2017; Barbeta et al. 2020). In addition, another recent glasshouse study showed that this isotopic offset was larger in plants forming symbiotic associations with mycorrhizal fungi (Poca et al., 2019)."

**RC: L62: seems a bit strange to refer to Poca et al. 2019 as an 'early study'?**

AC: We have modified the text in line with this suggestion which now reads "Another recent glasshouse study".

**RC: L116: it would be good to provide more information on the approach to combine LC-excess with SW-excess.**

AC: We have added a sentence to this paragraph to clarify this issue: "When we found that estimated values of both SW-excess and LC-excess for a given plant were significantly different from zero, this would indicate that there could be a genuine mismatch in isotopic composition between plant water and its most likely sources."

**RC: L146: here is a list of source waters. If I understand correctly though only soil water was directly tested. The other data were not even extracted, so this would be good to clarify here. I understand precipitation was analysed indirectly via LC-excess, but again the data were not used directly in this study.**

AC: We have removed from the text "precipitation, groundwater and streamflow" since these were not used in subsequent analyses.

**RC: Methodology:**

**It is not immediately clear what a 'sampling campaign' corresponds to. Is this a 'sampling occasion' at a specific site and within each plot? The word 'grouped' in line 168 might be why I'm confused, because if you group all dates and plots from a study site, it is unclear why you would have more campaigns than studies. Or does each study report on data from on average 5 study sites? It would be helpful to see the word 'sampling campaign' reappear as a heading in the right place in Table 1.**

AC: We have clarified the definition of campaign as follows: "For our analyses, we considered that the experimental unit was the sampling 'campaign'. We defined a campaign as a data collection event that occurred within a study site (or plot) within a limited time interval, thus each study consisted of one or more campaigns, depending on the number of sampling events." (L168). We have added a column to Table 1 indicating the number of 'campaigns' included per study and clarifying that this number is the number of plots times the number of sampling events, per study. Also, we included a new column for total data regarding the combination of species sampled in each campaign.

**RC: Soil water content is considered as an effect of climate, however, this is also a strong function of the soil properties. Climate might explain the main variations in VSM overtime at a specific site, but different soil types under the same climate can have vastly different absolute values at one moment in time. In this study, VSM is simply a variable across all campaigns and time locally is not considered (e.g. in the way it is plotted in Figure 4b). I'd therefore suggest rewording the role VSM represents (not simply to look at climate effects only).**

AC: We agree that soil volumetric water content (VWC) is not solely a function of climate, but also of soil properties. In section 2.3. of the Methodology we detail the collection of these and other data, the title of this section is "Climatic, environmental and biological data". We believe that the term "environmental data" is sufficiently broad enough to include soil VWC. We have carefully revised our text to make sure that soil VWC is not referred to as merely "climatic data".

**RC: Results:**

**Figure 2: the caption seems to be incorrect/unclear. I don't see n reported in the figure. The small numbers on the left-hand side appear to refer to the ID in table 1. This is helpful, but several study sites appear more than once. Therefore, the statement in the first few sentences referring to 'for each study site/sampling site' might be incorrect. I also find the think bars in the figure misleading. They suggest a range rather than a mean value, and a range that either starts or stops at 0. It would be better to have the mean as a coloured marker, also so that the SE is clearly equal distance either side of the mean.**

AC: We have changed this figure and clarified its caption according to this suggestion. The Y axis depicts the study sites and since some studies included more than one study site, that is why some numbers are repeated. The figure no longer depicts bars, but only the symbols with their corresponding error bars.

**RC: L307-312: this is very difficult to follow. Maybe split up or report the values in a table?**

AC: We have modified the text according to this suggestion: "The results of the LMMs including climatic and environmental variables in the fixed part of the model indicated that the slope of the SWL slope was sensitive to various climatic drivers. The slope of the SLW decreased with warmer temperatures (Table 2). In contrast, the slope of the SWL increased with soil VWC of the upper soil layers (Table 2) and with integrated soil water content (Table 2). Finally, the slope of the SWL also increased with annual and monthly precipitation (Table 2)".

**RC: L335-339 and Figure 4: I wonder why only Tair and Soil VWC for the upper 1 m are plotted in the figure? It would be good to see the data for PET as well and equally it would be interesting to see if the slope of the relationship for VWC of different soil depths changes and how?**

AC: In this revised version of our manuscript, we present the suggested graphs in the supplementary material (Figure S1). Yet, please note that our results indicated that upper soil VWC measured at 0-7, 7-28 and 28-100 cm depth had a significant negative effect on SW-excess and that the estimated effects for all these variables are presented in Table 2, together with the estimated effect of potential evapotranspiration (PET). We opted to only present in the main manuscript graphically the effects of monthly air temperature and integrated soil water content because we believe that these were the most informative ones.

**RC: Table 2: is it correct that 'estimate' refers to the 'slope estimate'? If yes, I suggest to edit the heading as such.**

AC: We have edited the caption of Table 2 which now reads: "Results (*t* and *P*), sample size (*n*) estimated slope and standard error (expect for the null models where 'Estimate' is the intercept) according to the linear mixed models.".

**RC: Discussion:**

**L405-411: this appears to be misplaced as the authors report additional analysis here (would fit better in the methods and results section)**

AC: We agree that this fragment reports the result of a particular analysis. This analysis is already reported in Results (L341):

Importantly, a more detailed analysis of the residuals of the relationship between SW-excess and the SWL parameters (slope and intercept) revealed that only the temperature effects had a direct effect on SW-excess (Table S3). On the other hand, the observed effects of soil VWC on SW-excess appeared to be a consequence of the direct effect of soil VWC on the SWL slope and intercept (Table S3).

However, we considered it important to repeat what the results in the analysis were? to improve the readability, i.e., to avoid that the readers have to go back and forth to find the results in another section. Still, we have slightly modified the sentence so it is more focused on the discussion of the results. The sentence reads as following:

Therefore, we ran the subsequent analysis of the residuals to that teased apart as direct and indirect environmental effects on SW-excess (Table S3). This analysis revealed that the negative effect of soil VWC on the SW-excess was mediated by the variability in the parameters of the SWL.

**RC: L 407: change 'run' to 'ran'**

AC: Typo edited accordingly, thank you.

**Literature cited**

**R1**

Barbeta, A., Burlett, R., Martin-Gómez, P., Fréjaville, B., Devert, N., Wingate, L., Domec, J.-C. and Ogée, J.: Evidence for distinct isotopic composition of sap and tissue water in tree stems: consequences for plant water source identification, bioRxiv, 2020.06.18.160002, doi:10.1101/2020.06.18.160002, 2020.

Benettin, P., Volkmann, T. H. M., von Freyberg, J., Frentress, J., Penna, D., Dawson, T. E., & Kirchner, J. W. Effects of climatic seasonality on the isotopic composition of evaporating soil waters. Hydrology and Earth System Sciences, 22(5), 2881–2890. doi:10.5194/hess-22-2881-2018, 2018.

Chen, Y., Helliker, B. R., Tang, X., Li, F., Zhou, Y. and Song, X.: Stem water cryogenic extraction biases estimation in deuterium isotope composition of plant source water, , doi:10.1073/pnas.2014422117, 2020.

Martinez-Vilalta, J., Anderegg, W. R. L., Sapes, G., & Sala, A. Greater focus on water pools may improve our ability to understand and anticipate drought-induced mortality in plants. New Phytologist, 223(1), 22–32. doi:10.1111/nph.15644, 2019.

[revised manuscript text omitted]

---

## Author Response (AR2)

In this document we provide a detailed point-by-point response to the comments raised by an anonymous reviewer and by the editor on our manuscript submitted to Hydrology and Earth System Sciences (hess-2021-333). Reviewer and editor comments are in **bold** font. In this document, line (L) numbers refer to the revised version.

**COMENTS BY ANONYMOUS REVEWER #3:**

**This manuscript presents a very interesting meta-analysis in relation to the commonly found (both in field and controlled conditions studies) offsets between precipitation, soil, and plant water stable isotope composition. It explores the potential biotic or abiotic reasons behind these offsets and exhaustively discuss them with current literature. I believe the methodology chosen by the authors (both LC-excess and SW-excess analysis) and statistical approach are correct and that they properly addressed and solved the comments done by previous reviewers. I have a few minor changes that I believe could improve at some points the manuscript:**

Thank you very much for these comments and for the time invested in reviewing our manuscript.

**Line 29. Rewrite lines 29-30-31. I would put together the sentence "SW-excess was more negative in cold and wet sites…" with the one in line 30 "The climatic effects…" so you can read the results together their implications.**

Thank you, we have modified the text accordingly (L31-32).

**Line 146. Did you check if there were differences associated to the type of "stem water" was collected? I understood in line 167 that you couldn't address the potential effects of water extraction methodology, but I am not sure you could analyze different type of stem samples.**

In our database 92 out of the 112 studies sampled exclusively woody species (shrubs and/or trees). In our database, we only included values of water isotopic composition obtained from samples of lignified tissues (i.e. wood cores or small twigs). For the remaining 20 studies focusing on non-woody species, only water samples from rhizomes, root collars or other similar tissues were considered. Hence, in our study all plant (stem) samples included are assumed to not have undergone any type of isotopic enrichment due to evaporation and we did not consider comparing among sampled organ types. Nonetheless, in our study the effect of "type of stem" would be implicit in the analyses of the effects of growth from (L170-171). Regarding differences in water isotopic composition (and their associated SW-excess and LC-excess) among stem water pools (mainly xylem water and storage water, following Barbeta et al. 2022), the paucity of studies using extraction methodologies alternative to cryogenic water distillation (three out of 112) allowing for the distinction between water pools currently impairs this comparison. Fortunately, in the near future, surely numerous studies applying the recent methodological advances we are witnessing in the field (Barbeta et al. 2022; Chen et al. 2021) will enrich the available data to address this question.

**Line 389. Lower/ higher. Always it is useful to add more negative, less negative…**

Thank you, we have modified the text accordingly (L390)

**Line 521. In this last paragraph I encourage the authors to include some discussion about plant hydraulic strategy and capacitance. Did you explore if there could be any effect associated to the fact that a species is more isohydric or anisohydric? Leaf hydraulic habits are generally associated to stem hydraulic patterns (anatomy, behavior). For example, as isohydric species close stomata quite fast in response to soil water potential changes, one could expect that they would avoid using "storage water", (for example the water in the parenchyma here assessed). We could expect the opposite behavior for anisohydric species and then higher differences between soil and stem water. It is just a hypothesis but, in my opinion, it would be worthy to explore. Indeed, in line 505, the**

**"hydraulic strategy" it is mentioned but I believe the authors only checked angiosperm vs. gymnosperms.**

These are very interesting remarks, also very relevant to our study interpretation. Indeed, the implications of storage water use are discussed in the main text (L436-437). Regarding the discussion on the "hydraulic strategy", we agree that our approach (comparison of gymnosperms vs. angiosperms, L505-507) constitutes a coarse conceptual simplification and that within these clades there is actually a whole gradient of hydraulic strategies implying the coordination of stomatal behaviour, leaf and wood anatomical traits (e.g. Rosas et al. 2019). On the other hand, the distinction between isohydric and anisohydric species can also be viewed as a continuum and it has been shown that a tight stomatal regulation might not be necessary linked to, for example, vulnerability to drought (Martínez-Vilalta & García-Forner, 2017). Even more, there are species that can depict significant plasticity in their stomatal behaviour (e.g. Guo et al. 2019, Wu et al. 2021), making the distinction between isohydric and anisohydric species even more blurry.

**COMENTS BY THE EDITOR (Prof. Dr Markus Weiler):**

**1) Were the SWLs and LMWLs calculated by orthogonal least squares fitting? They should be because both the X and Y data have uncertainty, and fitting by standard least squares can artificially reduce slopes, which could have consequences for the findings. -> you provided a Figure showing that the slope of a regression line is not statistically different from 1. This is true, but since the correlation is quite low, there is no wonder, that the line is not different from one. However, if you would look at the differences of individually calculated SWL by OLS vs TLS, the can be different up to 5 per mil. The question is, does this matter in the further analysis. So please convince me, that the results in the paper using the SWL is the same if you use OLS or TLS is the same**

For the studies included in our meta-analysis, we found that when fitting our soil water lines (SWL's) for datasets where there was a strong linear correlation between $\delta^2H$ and $\delta^{18}O$ of soil water, there were no differences in the slope or intercept term of the SWL between the ordinary (or total) least squares and the orthogonal least squares methods (see bottom panels of Figure A in this document for some examples). In contrast, for those studies where there were a few data points where the relationship between $\delta^2H$ and $\delta^{18}O$ was clearly different from the rest of the data (see for example the graphs corresponding to Goldsmith et al. 2012 or Gómez-Navarro et al. 2019 in Figure A in this document), the orthogonal least squares method underestimated the intercept and overestimated the slope of the SWL, with respect to estimates from the ordinary least squares. In such cases, estimation of the SWL parameters with the orthogonal least squares method would have resulted into an overestimation of the absolute values of SW-excess. Wehr & Saleska (2017) noted that the orthogonal least square fitting is unbiased only when the error variances of the two variables ($\delta^2H$ and $\delta^{18}O$ in our case) are equal, but the error of $\delta^{18}O$ is usually smaller than that of $\delta^2H$. We argue that the ordinary least squares should be a good approximation for fitting our SWL's, as it should render much less biased estimates of the intercept than the orthogonal method (according to Wehr and Saleska 2017 and also to Zobitz et al. 2006)

[Figure]

**Figure A.** Dual plots of soil water isotopic composition ($\delta^2$H and $\delta^{18}$O) from a subset of studies included in the meta-analyses (see Table 1 and list of references in the main text for the corresponding references). The black lines depict the soil water lines (SWL) fitted with the ordinary least squares fitting and the green line depicts the SWL fit with the orthogonal least squares method.

**2) I agree with reviewer #2, that "My main concern relates to the approach towards the soil water data. Most studies used in the meta-analyses will have taken soil samples across a range of different depths.". In addition, most studies do not even sample the full range of rooting depth, hence they may miss pools that plants can take up. You argued, that there is no way of dealing with this, as you may not have the data and information to look at shallow or deep separately. But at least, you should provide a clear statement in the abstract and conclusion, that this is a main weakness of this meat-study, that the isotopic composition is sampled at different depths (and with different methods), which may be even more problematic.**

In agreement with your remark and those by reviewer #2, we found that there was large variability in sampling methodologies, depths and intervals. Following this suggestion, this is now explicitly stated in the abstract (L23) and in the conclusions (L542). Obviously, we cannot be sure that all potentially relevant water sources and soil layers were sampled in all studies, but this is an inherent limitation to any study using analyses of water isotopic composition to infer plant water sources. During our data collection, we recorded the depth of soil sampling for analyses of water isotopic composition. This information will be provided on the database associated to the publication, but we opted not to calculate SW-excess separately for deep and shallow soils because these were largely indistinguishable (see Figure

B in this document for some examples) and separating soil pools increased the uncertainty of the SWL fit.

[Figure]

**Figure B.** Dual plots of soil water isotopic composition ($\delta^2$H and $\delta^{18}$O) for deep (orange lines and symbols, depth $\geq$ 30 cm) and shallow (cyan lines and symbols, depth < 30 cm) samples from a subset of studies included in the meta-analyses (see Table 1 and list of references in the main text for the corresponding references). From the 112 studies, 92 of them reported sampling depths at shallow and deep layers of the soil profile.

**REFERENCES**

Barbeta, A., Burlett, R., Martin-Gómez, P., Fréjaville, B., Devert, N., Wingate, L., Domec, J.-C. and Ogée, J.: Evidence for distinct isotopic composition of sap and tissue water in tree stems: consequences for plant water source identification,New Phytol, 223(3), 1121-1132, doi:10.1101/2020.06.18.160002, 2022.

Chen, G., Li, X., Qin, W., Lei, N., Sun, L. Z., Cao, L. and Tang, X.: Isotopic fractionation induced by a surface effect influences the estimation of the hydrological process of topsoil, Hydrol. Process., 35(1), doi:10.1002/hyp.14019, 2021.

Guo, J. S., Hultine, K. R., Koch, G. W., Kropp, H., & Ogle, K. Temporal shifts in iso/anisohydry revealed from daily observations of plant water potential in a dominant desert shrub. *New Phytologist*, *225*(2), 713–726. https://doi.org/10.1111/nph.16196, 2019.

Martínez-Vilalta, J., & Garcia-Forner, N. Water potential regulation, stomatal behaviour and hydraulic transport under drought: deconstructing the iso/anisohydric concept. *Plant, Cell & Environment*, *40*(6), 962–976. https://doi.org/10.1111/pce.12846, 2016

Rosas, T., Mencuccini, M., Barba, J., Cochard, H., Saura-Mas, S., & Martínez-Vilalta, J. Adjustments and coordination of hydraulic, leaf and stem traits along a water availability gradient. *New Phytologist*, *223*(2), 632–646. https://doi.org/10.1111/nph.15684, 2019.

Wehr, R. and Saleska, S. R.: The long-solved problem of the best-fit straight line: application to isotopic mixing lines, Biogeosciences, 14, 17–29, https://doi.org/10.5194/bg-14-17-2017, 2017.

Wu, G., Guan, K., Li, Y., Novick, K. A., Feng, X., McDowell, N. G., Konings, A. G., Thompson, S. E., Kimball, J. S., de Kauwe, M. G., Ainsworth, E. A., & Jiang, C. Interannual variability of ecosystem iso/anisohydry is regulated by environmental dryness. *New Phytologist*, *229*(5), 2562–2575. https://doi.org/10.1111/nph.17040, 2020.

Zobitz, J. M., Keener, J. P., Schnyder, H., and Bowling, D. R.: Sensitivity analysis and quantification of uncertainty for isotopic mixing relationships in carbon cycle research, Agricultural and Forest Meteorology, 136, 56–75, https://doi.org/10.1016/j.agrformet.2006.01.003, 2006.